

# Land-atmosphere interactions in the tropics
Pierre Gentine
*Department of Earth and Environmental Engineering,*
*Earth Institute*
*Columbia University, New York, USA*
Adam Massmann
*Department of Earth and Environmental Engineering,*
*Earth Institute,*
*Columbia University, New York, USA*
Benjamin R. Lintner
*Department of Environmental Sciences*
*Rutgers, The State University of New Jersey*
*New Brunswick, NJ, USA*
Sayed Hamed Alemohammad
*Department of Earth and Environmental Engineering,*
*Earth Institute,*
*Columbia University, New York, USA*
Rong Fu
*Atmospheric and Ocean Sciences Department*
*University of California, Los Angeles*



28 Julia K. Green

29 *Department of Earth and Environmental Engineering,*

30 *Earth Institute,*

31 *Columbia University, New York, USA*

33 Daniel Kennedy

34 *Department of Earth and Environmental Engineering,*

35 *Earth Institute,*

36 *Columbia University, New York, USA*

38 Jordi Vilà-Guerau de Arellano

39 *Meteorology and Air Quality Group*

40 *Wageningen University, Wageningen, the Netherlands*

46 _______________________

48 *Corresponding author address:* Pierre Gentine, Department of Earth and Environmental

49 Engineering, Columbia University, NY 10027, USA.

50 E-mail: pg23288@columbia.edu

51 Phone number: +1-212-854-7287



ABSTRACT
The continental tropics play a leading role in the terrestrial water and carbon cycles. Land-
atmosphere interactions are integral in the regulation of surface energy, water and carbon fluxes
across multiple spatial and temporal scales over tropical continents. We review here some of the
important characteristics of tropical continental climates and how land-atmosphere interactions
regulate them. Along with a wide range of climates, the tropics manifest a diverse array of land-
atmosphere interactions. Broadly speaking, in tropical rainforests, light and energy are typically
more limiting than precipitation and water supply for photosynthesis and evapotranspiration;
whereas in savanna and semi-arid regions water is the critical regulator of surface fluxes and
land-atmosphere interactions.  We discuss the impact of the land surface, how it affects shallow
clouds and how these clouds can feedback to the surface by modulating surface radiation. Some
results from recent research suggest that shallow clouds may be especially critical to land-
atmosphere interactions as these regulate the energy budget and moisture transport to the lower
troposphere, which in turn affects deep convection. On the other hand, the impact of land surface
conditions on deep convection appear to occur over larger, non-local, scales and might be
critically affected by transitional regions between the climatologically dry and wet tropics.

## 1   Introduction

Tropical ecosystems play a substantial role in regulating the global carbon and hydrologic
cycles. Tropical rainforests are one of the main terrestrial carbon sinks [*Nakicenovic*,
2000] but their projected response to a warming climate remains unclear because of
uncertainties associated with the representation of abiotic and biotic processes in models
as well as confounding factors such as deforestation and changes in land use and land
cover [*Wang et al.*, 2009; *Davidson et al.*, 2012; *Fu et al.*, 2013; *Saatchi et al.*, 2013;
*Hilker et al.*, 2014; *Boisier et al.*, 2015; *Doughty et al.*, 2015; *Gatti et al.*, 2015; *Knox et*
*al.*, 2015; *Saleska et al.*, 2016]. The ecosystems of tropical monsoonal and seasonal wet-
dry climates are also important contributors to the global carbon cycle, especially with
respect to the interannual variability of the tropical terrestrial carbon sink [*Poulter et al.*,
2014; *Jung et al.*, 2017].
Some regions of the tropics have been further identified as hotspots of land-atmosphere
interactions, modifying the regional climate [*Green et al.*, 2017] either locally, i.e. at
horizontal scales on the order of a few boundary layer heights, regionally, at scales up to



a few hundreds of kilometers, or at large scales, over several of thousands of kilometers,
through coupling between the surface and the overlying atmosphere [*Lintner and Neelin*,
2009]. While tropical land-atmosphere interactions are often examined through the lens
of coupling between land surface states (e.g., soil moisture) and rainfall, other aspects of
the coupling are also important. For example, even under nonprecipitating conditions,
surface radiation, temperature and vapor pressure deficit (VPD) may be altered [*Lawton*
*et al.*, 2001; *Pielke et al.*, 2016; *Green et al.*, 2017] through coupling with clouds,
aerosols and shallow (non-precipitating) convection [*Avissar and Nobre*, 2002; *Medvigy*
*et al.*, 2011; *Seneviratne*, 2013; *Cook et al.*, 2014; *Guillod et al.*, 2015; *Krakauer et al.*,
2016; *Martin et al.*, 2016; *Green et al.*, 2017; *Khanna et al.*, 2017; *Martin et al.*, 2017;
*Thiery et al.*, 2017; *Vogel et al.*, 2017].  It is clear that the tropical energy, water, and
carbon cycles cannot be understood in isolation; rather, interactions among these cycles
are critical, especially in determining whether the terrestrial tropics will act as a future
carbon sink or source [*Zhang et al.*, 2015][*Swann et al.*, 2015].
The two-way interactions that occur between the land surface and overlying atmosphere
represent one of the more uncertain aspects of the terrestrial climate system, particularly
in the tropics [*Betts and Silva Dias*, 2010]. While the land surface is widely recognized as
integral to the occurrence of important tropical climate phenomena such as monsoons
[*Zeng and Neelin*, 1999; *Zeng et al.*, 1999], isolating and quantifying its precise role
remains elusive. Indeed, such efforts have frequently been hampered by the paucity of
observational data, not to mention the complex and multiple pathways through which
land-atmosphere interactions can take place.
Several field campaigns have been conducted in the tropics with the purpose of
advancing knowledge of land-atmosphere interactions. One of the first campaigns was
the Large-Scale Biosphere-Atmosphere Experiment in Amazonia (LBA) [*Avissar et al.*,
2002; *Keller et al.*, 2004], which aimed at refining our understanding of climatological,
ecological, biogeochemical and hydrological processes of the Amazon and their linkages,
in addition to the anthropogenic impacts (e.g., land-use land cover changes and
deforestation, in particular) on these.  Among many other topics, LBA generated
fundamental insights on the structure of the tropical atmosphere, processes generating
precipitation, and the seasonal variability of rainforest surface turbulent fluxes [*Avissar*
*and Nobre*, 2002; *Betts et al.*, 2002; *Laurent et al.*, 2002; *Machado and Laurent*, 2002;





*Acevedo et al.*, 2004; *Khairoutdinov and Randall*, 2006; *Fitzjarrald*, 2007; *Juárez et al.*,
2007; *Restrepo-Coupe et al.*, 2013]. Much of the initial LBA research attempted to isolate
the effect of deforestation on precipitation, both in a local context as well as remotely via
teleconnections [*Avissar et al.,* 2002]. Much of this research pointed to deforestation
decreasing precipitation, albeit with uncertain magnitude. Even now, two decades after
the inception of LBA, the relationship between tropical deforestation and precipitation
remains uncertain, despite progress with respect to key processes such as the forest's role
in accessing deep water in the dry season, and cloud-cover's role in modulating energy
availability for photosynthesis [*Betts and Dias*, 2010].
Another noteworthy field campaign, the African Monsoon Multidisciplinary Analysis
(AMMA) campaign, focused on the West African monsoon system, especially the Sahel
transition zone [*Redelsperger et al.*, 2006; *Boone et al.*, 2009b]. AMMA built upon
previous field work in the region [*e.g. HAPEX-Sahel*, *Gourtorbe et al. 1993*], and
substantially advanced understanding of mesoscale convective systems and their
initiation the role of surface profeses, and the vegetation water stress response in semi-
arid regions [*Lebel et al.*, 2009; *Taylor et al.*, 2009; *Boone et al.*, 2009a; *Lohou et al.*,
2010; *Couvreux et al.*, 2011a; 2011b]. More recently, the 2014-2015 Green Ocean
Amazon (GO-Amazon) campaign [*Martin et al.*, 2016] sought to quantify the impact of
atmospheric composition and aerosols under clean and polluted conditions on cloud
formation and radiation over the basin, as well as on shallow to deep convection
development [*Anber et al.*, 2015a; *Tang et al.*, 2016; *Giangrande et al.*, 2017].
The remainder of this review article is organized as follows. We first review the typical
definitions of the tropics and of land-atmosphere interactions in section 2. In section 3 we
discuss the seasonality and characteristics of the climate of the tropics. The different
types of feedbacks from local to non-local (i.e. remote influences) are then highlighted in
section 4, and we close in arguing that shallow cloud feedback and its impact on radiation
has received too little attention compared to precipitation feedback, in rainforests
especially.





## 2   Definitions of land-atmosphere interactions in the tropics

### 2.1   What (where) are the Tropics?

There exist multiple definitions of the Tropics. On the one hand, the Tropics can be defined spatially as the area between the Tropics of Cancer and Capricorn, located at ~23½° N and ~23½° S, respectively. On the other hand, it is sometimes useful to define the Tropics in terms of underlying climate or physical characteristics. One such physically-motivated definition of the Tropics is the region over which mean top-of-the-atmosphere solar incoming radiation exceeds outgoing radiation (reflected shortwave and outgoing longwave), which occurs equatorward of ~35°. Another definition is the region near the equator where the Coriolis effect is small and planetary scale equatorial wave dynamics are dominant, which strongly affects the dynamics, as we elaborate below [*Sobel et al.*, 2001; *Sobel and Bretherton*, 2003; *Lintner and Chiang*, 2005; *Raymond and Zeng*, 2005].

Over land, the Tropics are often defined biogeographically, as in the traditional Köppen climate classification scheme [*Köppen*, 1884]: tropical regions are divided into three main groups—tropical rainforest, tropical monsoon, and tropical wet and dry (or savanna)—all of which are characterized by annual mean temperatures exceeding 18°C but which differ in terms of precipitation amount and seasonality.

The latitudes between the Tropics of Cancer and Capricorn encompass some regions of large-scale subsidence and limited rainfall, including drylands and deserts, which we largely neglect here, even though land-atmosphere coupling processes within these regions is clearly of interest. Thus, throughout this manuscript, we define the Tropics as the latitudinal band between -15° S and 15° N, as it captures most of the wet regions of the Tropics while excluding many of the more arid regions at higher latitudes.

### 2.2   How to define land-atmosphere interactions?

There are typically two main definitions of land-atmosphere interactions:

#### 2.2.1   Surface turbulent fluxes

While many potential definitions of land-atmosphere interactions exist, we propose a definition of land-atmosphere interactions as the study of turbulent fluxes and associated momentum, energy, water and trace gases exchanges between the biosphere and the



atmosphere [*Goulden et al.*, 2004; *Fisher et al.*, 2009; *Restrepo-Coupe et al.*, 2013].
Surface turbulent flux measurements in the tropics are usually obtained from eddy-
covariance methods, typically above the canopy [*Baldocchi et al.*, 2001]. Observing
turbulent fluxes is challenging in tropical environments for many reasons including
logistics, maintenance and the harsh environment such as intense rainfall, high wind, and
relative humidity, which impacts the sensors [*Campos et al.*, 2009; *Da Rocha et al.*,
2009; *Restrepo-Coupe et al.*, 2013; *Zahn et al.*, 2016; *Chor et al.*, 2017; *Gerken et al.*,
2017]. In light of these challenges, it is perhaps not surprising that even the best estimates
of surface turbulent fluxes manifest large uncertainties [*Mueller et al.*, 2011].
Apart from site level measurements, remote sensing observations can provide
information about surface turbulent fluxes and other relevant quantities over tropical land
regions. There is considerable uncertainty in upscaling point observations to larger areas.
Remote sensing observations are useful to generalize and compare fluxes across the
tropics even if they are not as direct as point observations, which are limited to ~ 10 local
stations across the wet tropics. We emphasize that there are considerable uncertainties in
remote sensing and reanalysis estimates of rainfall [*Washington et al.*, 2013; *Levy et al.*,
2017], radiation [*Jimenez et al.*, 2011], and surface turbulent fluxes [*Alemohammad et al.*,
2016].
While direct, satellite-based retrievals of turbulent fluxes of carbon (i.e. gross primary
production (GPP))) and water would be most suitable for the study of tropical land-
atmosphere interactions, such retrievals are beyond current remote sensing capability.
However, some recent work demonstrates that existing satellite observations may still be
leveraged to study surface turbulent fluxes in the tropics. *Alemohammad et al.* [2016]
applied a machine learning algorithm based on remotely-sensed Solar-Induced
Fluorescence (SIF), called WECANN (Water Energy and Carbon Artificial Neural
Network) to derive surface turbulent fluxes.  WECANN reproduces the seasonality in the
wet tropics and exhibits plausible interannual. In contrast to the normalized difference
vegetation index (NDVI) or many other vegetation indices which are indirect byproducts
of photosynthesis, SIF (at the leaf scale) is directly related to the ecosystem-scale
photosynthesis rate, providing important information on the impact of stressors on
photosynthesis and is available from existing remote sensing platforms [*Frankenberg et*
*al.*, 2011; *Joiner et al.*, 2011; *Frankenberg et al.*, 2012; *Joiner et al.*, 2013; *Frankenberg*



*et al.*, 2014; *Guanter et al.*, 2014; *Lee et al.*, 2015; *Duveiller and Cescatti*, 2016; *Liu et*
*al.*, 2017; *Thum et al.*, 2017; *Alexander et al.*, n.d.]. SIF is thus an important indicator of
the rates of photosynthesis and transpiration through stomatal (small pores at the leaf
surface) opening [*Alemohammad et al.*, 2017]. Indeed, during photosynthesis plants take
up $CO_2$ from the atmosphere while releasing water to the atmosphere through stomata.
WECANN performs well compared to eddy-covariance observations and has less
uncertainty compared to many other retrievals (see [*Alemohammad et al.*, 2017]). We
note that recent developments in observations of SIF seem to indicate that the major
fraction of the SIF signal might be related to chlorophyll photosynthetically active
radiation and that changes in SIF yield (equivalent to light use efficiency) may account
for only a small fraction of the observed SIF signal [*Du et al.*, 2017]. This is still an open
topic to better understand what is actually observed by SIF remote sensing.
### 2.2.2   Weather and climate feedback
A second definition of land-atmosphere interactions relates to the feedback between
surface processes (radiation, surface turbulent fluxes) and the overlying atmosphere,
which may occur across multiple temporal and spatial scales. Throughout this
manuscript, we highlight contribution of three types of feedbacks:
1) feedbacks between the surface and low-level clouds, including surface fog and shallow
convection;
2) feedbacks between the surface and deep convection, i.e. deep raining clouds extending
above the freezing level;
3) feedbacks between the surface and large-scale circulation.
The distinction between shallow and deep convection remains elusive, as these have been
regarded as both fundamentally distinct or as a continuum, in both observations and
model convection parameterizations [*Khairoutdinov and Randall*, 2006; *Bretherton and*
*Park*, 2009; *Park and Bretherton*, 2009; *Rio et al.*, 2009; *Wu et al.*, 2009; *Del Genio and*
*Wu*, 2010; *Hohenegger and Bretherton*, 2011; *Böing et al.*, 2012; *D'Andrea et al.*, 2014;
*Rochetin et al.*, 2014b]. We will loosely refer to shallow convection as convection
confined below the freezing level (typically less than 3km deep) and comprising non-
precipitating clouds with motions of small scale (typically less than a km in the
horizontal).



An important point is that shallow convection is frequently generated by thermals rooted
in the boundary layer and is thus ultimately related to surface sensible (H) and latent heat
(LE) flux and their partitioning [*Gentine et al.*, 2013a; 2013b; *de Arellano et al.*, 2014].
The impact of surface heat fluxes and their partitioning on shallow convection is
demonstrated in the Amazon in **Figure 1**. Shallow convection frequently occurs over the
vegetated surface away from the ocean; also, over cooler and more humid river basins,
shallow clouds are virtually absent [*Gentine et al.*, 2013a; *Rieck et al.*, 2014; 2015]. In
addition, shallow convection is strongly influenced by the diurnal cycle of surface
radiation and surface turbulent heat fluxes [*Gentine et al.*, 2013a; 2013b; *de Arellano et
al.*, 2014].
On the other hand, we use the term deep convection in association with deep,
precipitating clouds. Deep convection may be triggered by boundary layer thermals
[*D'Andrea et al.*, 2014; *Guillod et al.*, 2014; *Rochetin et al.*, 2014a; 2014b; *Anber et al.*,
2015a] as well as other processes such as radiative destabilization [*Anber et al.*, 2015b],
meso- and large-scale circulations [*Werth and Avissar*, 2002; *Roy et al.*, 2003], cold pools
(cold density currents due to rain evaporation that cools the air within precipitating
downdrafts) [*Engerer et al.*, 2008; *Del Genio and Wu*, 2010; *Böing et al.*, 2012; *Feng et
al.*, 2015; *Torri et al.*, 2015; *Gentine et al.*, 2016; *Heever*, 2016; *Drager and van den
Heever*, 2017] and wave activity [*Kuang*, 2008; 2010]. As such, deep convection may be
viewed as less dependent on the surface state compared to shallow convection.
Over the central Amazon a large fraction of wet season precipitation occurs during the
nighttime (Figure 2). Moreover, during the daytime in both the dry and the wet seasons,
the diurnal cycle reflects not only locally surface-triggered deep convection
[*Khairoutdinov and Randall*, 2006; *Ghate and Kollias*, 2016] but also mesoscale
convective systems propagating on daily time scales throughout the Amazon basin
[*Ghate and Kollias*, 2016]. However, during the dry season, precipitation occurs more
frequently with the "popcorn type" deep convection that is more locally triggered and
thus directly related to the state of the land surface [*Ghate and Kollias*, 2016] (see an
example here https://youtu.be/c2-iquZziPU).
Current generation climate models struggle to represent both shallow and deep
convection over continents [*Guichard et al.*, 2004; *Bechtold et al.*, 2013; *Yin et al.*, 2013;
*D'Andrea et al.*, 2014; *Couvreux et al.*, 2015], and especially in the tropics, as they



exhibit substantial errors in the phasing and intensity of both the diurnal and seasonal
cycles of convection [*Bechtold et al.*, 2013], as well as biases in the climatological
distribution of rainfall over land. For example, over the Amazon, many climate models
underestimate surface precipitation, evapotranspiration, and specific humidity [*Yin et al.*,
2013], with the dry bias in moisture extending upwards into the lower free troposphere
{Lintner:2017gm}. Such biases are largely thought to reflect deficiencies or errors in how
convection is represented in models [*Yano and Plant*, 2012; *Stevens and Bony*, 2013;
*Bechtold et al.*, 2014]. Indeed, in current generation climate models, cloud processes
occur at scales smaller than resolved grid-scale prognostic variables and therefore need to
be parameterized, i.e. represented as a function of the resolved-scale variables. This is
important as it means that climate models do not explicitly represent the small-scale
convective physics of the climate system. We do note, however, that cloud resolving
models which include explicit convection at scales of ~1km alleviate many of the biases
observed in climate models, especially in terms of the diurnal cycle of convection or the
sign and magnitude of the feedbacks between deep convection and surface evaporative
fraction [*Taylor et al.*, 2013; *Anber et al.*, 2015a]. Nonetheless, due to convective wave
coupling in the Tropics, a simple prescription of lateral boundary conditions in small-
domain cloud-resolving model may be problematic, as the convective scales ultimately
interact and are coupled with the planetary scales. With a sufficiently large domain and
fine enough resolution, coupling between the convective scales and planetary scales may
be explicitly resolved, but simulations of this nature are likely too be computationally too
expensive for many applications. However, techniques exist to represent the effect of
large-scale dynamics on the convective scales, which, when combined with cloud
resolving simulations, yield powerful tools for understanding land-atmosphere
interactions in the tropics, as we elaborate further below.
## 3   Characteristics of the tropics
### 3.1   Weak temperature gradient approximation – nonlocality
One key concept in tropical climate is the Weak Temperature Gradient (WTG)
approximation.  In the tropical free troposphere, horizontal gradients of temperature (and
pressure) are small in part because of the relative weakness of the Coriolis parameter (as





on large-scales, geostrophic balance holds poleward of ~5 degrees). Homogenization
occurs over a spatial scale comparable to the Rossby radius of deformation, which is
inversely proportional to the Coriolis parameter. In midlatitudes, the Rossby radius is of
order $10^2$ km (similar to climate model resolution). In the tropics, the Rossby radius is
typically an order of magnitude larger. Consequently, localized convection, and the
diabatic heating associated with condensation and freezing of water, cannot be viewed in
isolation from the large-scale in the tropics: in other words, in the tropical free
troposphere, the temperature and pressure fields rapidly adjust to localized perturbations,
effectively spreading the effect of these perturbations. In addition, it is relatively
straightforward to show that adiabatic cooling, associated with large-scale vertical ascent
in the presence of a vertical gradient of dry static energy $h = c_p T + gz$, effectively
balances the diabatic heating rate **Q**, which in rainy regions of the tropics is mostly
associated with convective processes. This further emphasizes the coupling between
diabatic heating and large-scale ascent. Since the introduction of WTG, related and
refined frameworks, such as weak pressure gradient [*Romps*, 2012a; 2012b] or damped
gravity waves [*Wang et al.*, 2013], have been proposed. It should be emphasized that the
WTG framework is only valid in the free troposphere, above the boundary layer, as it
relates to wave dynamics in a stratified atmosphere.
The WTG framework has been used in single-column model and cloud-resolving models
of the tropics [*Sobel et al.*, 2007; *Daleu et al.*, 2012; 2014; *Sentić and Sessions*, 2017] to
obtain boundary conditions consistent with convective activity in the domain, thus
avoiding the issues of inconsistent boundary forcing alluded to in section 2.2.2. While the
WTG framework has often been applied in an oceanic context, [*Anber et al.*, 2015a] have
demonstrated its utility in studying the coupling between regional land surface processes
and larger-scale circulation, as discussed in Section XX.

### 3.2    Surface turbulent fluxes climatology and seasonality

Given that few flux towers are available across the tropics, we use WECANN
[*Alemohammad et al.*, 2017] to calculate surface flux climatologies across the continental
tropics. WECANN has been validated against available flux tower data and outperforms
other products in terms of reproducing both the seasonality and interannual variability
[*Alemohammad et al.*, 2017]. While remote sensing retrievals are not perfect and cannot
be considered the truth, they do provide spatially extensive data coverage, including



regions with sparse (or no) site-level measurements (e.g., Congo), which are hard to
upscale to larger scale.  In what follows, we evaluate climatologies of evapotranspiration
(ET) and gross primary production (GPP) against precipitation (based on GPCP 1DD
v1.2 [*Huffman et al.*, 2001]) and net radiation (based on CERES SYN [*Kato et al.*, 2013])
(Figure 4 to Figure 8).
We first focus on the main tropical rainforests and the northeastern savanna (or Cerrado)
region of Brazil (Figure 4). In the wetter part of the Amazon, net radiation, Rn, peaks in
the dry season (August to November) (Figure 4) when precipitation and cloud cover—
especially shallow cloud cover, including fog—are reduced, [*Anber et al.*, 2015a]. As a
result of the reduced cloud cover, incident surface solar radiation increases, and both GPP
(Figure 6) and ET (Figure 7) increase in the dry season (Figure 4).  As discussed further
in the next section, the forest in the climatologically wetter Amazon is primarily light
limited, while water stress there is moderate in the dry season. The seasonal cycle is more
pronounced for GPP than for ET (Figure 4), as canopy rain interception comprises a large
fraction of total ET in the wet season [*Scott et al.*, 1997; *Oleson et al.*, 2008; *Miralles et*
*al.*, 2010; *Sutanto et al.*, 2012; *van Dijk et al.*, 2015; *Andreasen et al.*, 2016] and partly
compensates for reduced transpiration in the wet season. In fact, because of this
compensation, the wettest parts of the Amazon exhibit weak ET seasonality. On the other
hand, most land-surface models exaggerate water stress in the Amazon [*Powell et al.*,
2013] and typically exhibit much lower rates of ET and GPP in the dry season, as well as
opposite seasonality of net ecosystem exchange, than are observed [*de Gonçalves et al.*,
2013; *Alemohammad et al.*, 2016; 2017].
In contrast to the everwet central Amazon, over the Cerrado region of Northeastern
Brazil, the seasonal cycles of Rn, precipitation, GPP and ET are much more pronounced,
with a marked dry season (Figure 4). The seasonal cycle of GPP tracks precipitation,
exhibiting a strong increase during the wet season. Similarly, ET increases sharply in the
wet season and then decreases more slowly than precipitation in the dry region (Figure
4). Conversely, net radiation increases sharply during the dry season.  This region clearly
exhibits a strong water stress response.
Over the Maritime Continent, rainfall is intense throughout the year and seasonality is
modest, with a short peak in November to January (Figure 4). Much of the seasonal cycle
is attributable to monsoon circulations, which are strongly influenced by topography and



the land- and ocean-surface thermal contrast [Chang 2005]; however, the complexity of
the topography and the distribution of island land masses leads to strong local variability.
Additionally, the Madden Julian Oscillation, an important mode of climate variability in
in the tropical Indo-Pacific with a lifecycle of 30-90 days, strongly impacts rainfall on
intraseasonal timescales [Hidayat and Kizu, 2009]. Convective activity in the region also
regulates the East Asian Monsoon [*Huang and Sun*, 1992]. The region is also influenced
by topographic effects and land-see breeze interactions at shorter time scales, and
exhibits a strong diurnal cycle in convection [*Nitta*, 1987; *Hamada et al.*, 2008]. Given
the relatively steady annual cycle of precipitation with regular convection, ET and GPP
remain relatively steady throughout the entire year, exhibiting minimal seasonality, in
this light limited environment (Figure 4).
The Congo basin exhibits two rainy seasons (Figure 4), with peaks in March-April-May
and September-October-November, related to seasonal changes in moisture convergence
due to the African Easterly jet and Intertropical Convergence Zone (ITCZ) over the
Atlantic [*Washington et al.*, 2013]. Throughout the year, monthly-mean precipitation is
much less than that observed over the Amazon or Indonesia.  The seasonality of GPP and
ET, to a lesser extent, tracks that of precipitation, with substantial decreases during the
June to August dry season and even more pronounced reduction during the December to
February period. This seasonality in GPP and ET (Figure 4) suggests that the Congo
basin should exhibit substantially more water stress during dry seasons compared to the
Amazon or Indonesian rainforests (Guan et al. 2015).
Integrated over the entire tropical latitudinal band, precipitation is highest in DJF and
MAM when the wet season extends over most of the Amazon and adjacent savanna
regions (Figure 5). GPP is maximized during the wet season in South America, as GPP is
highest in the savanna regions while GPP over the rainforest is effectively seasonally
invariant (Figure 7). The seasonal pattern of ET resembles GPP (Figure 7), mostly
reflecting the seasonality of water availability in drier, water-limited regions and
increased radiation in the dry season in the wetter, more energy-limited portions of the
Amazon. The seasonal cycle of sensible heat flux (Figure 8) largely follows water stress,
especially in the rainforest where radiation remains high throughout the year, with an
increase during the dry season. Water stress is further evidenced in the evaporative
fraction, EF, the ratio of latent heat flux to latent and sensible heat fluxes (Figure 9).
Tropically-averaged EF does not evolve much reflecting seasonal variation in the
latitudinal peak in radiation and compensation of decreased canopy interception by
transpiration (because of increased net surface radiation) in the dry season. However, in
transitional and dry regions to the east, EF exhibits substantial seasonal variation between
the wet season, when it peaks, and the dry season. The surface moist static energy flux
(assuming sea level elevation) shows variations in SON and JJA but otherwise remains
steady across longitudes because of compensation between the increased H and reduced
ET. In the dry to wet transition, SON, moist static energy flux exhibits an interesting peak
at about -60 longitude (Figure 10) though the combined increase in radiation, due to
reduced cloudiness, inducing higher sensible heat flux and maintained high ET rates.
Over tropical Africa, the precipitation is highest in JJAS during the wet phase of the West
African Monsoon, with a secondary maximum in DJF corresponding to the Southern
African Monsoon (Figure 5). Similarly the latitudinal-averaged GPP and ET increase
during the West African Monsoon (Figure 6, Figure 7), accompanied by a strong
decrease in sensible heat flux (Figure 8). In DJF the southern African Monsoon displays
increased water flux (Figure 7) and photosynthesis tracking the increased rainfall (Figure
5). The Congo rainforest clearly exhibits two brief rainy seasons (Figure 4, Figure 9),
with peaks in March-April-May and September-October-November (Figure 4) and
displays substantial water stress and strong reduction in EF to values below 0.6 during
the dry season (Figure 9).
### 3.3 Rainforest water stress
One outstanding challenge in modeling tropical land regions is why do most
contemporary land-surface models incorrectly represent the wettest rainforest GPP and
ET rates, their seasonal cycles, and how they relate to water stress? Capturing this
accurately will help better understand the seasonal course of GPP and ET in the tropics.
In the wettest tropical forests, such as the western portion of the Amazon or Indonesia,
energy and light limit the rates of ET and GPP. It is thus natural to conclude that soil
moisture and water stress have only minor effects in such regions and thus that
precipitation variability would not matter much. In fact, there exist sharp vertical
gradients in the canopy (as well as at the surface of the soil in the dry season) in terms of
light and water availability (along with nutrient allocation) (Figure 3). Understory species
receive only a small amount of mostly diffuse light. However, water is not typically





limiting for low-canopy species. Moreover, because relative humidity is high and VPD is
low, leading to low stress on understory stomatal and ecosystem conductance [*Leuning*,
1995; *Leuning et al.*, 1995; *Wang and Leuning*, 1998; *Medlyn et al.*, 2011; 2012; *Heroult*
*et al.*, 2013].
On the other hand, top canopy species receive a large amount of radiation, especially in
the dry season, causing sunlit leaf warming and desiccation leading to heat and water
stress [*Jardine et al.*, 2014]. Leaf and xylem water status are regulated by the relative
demand of sap from transpiration, which depends on incoming radiation, temperature and
VPD. It also depends on the supply of sap to the leaves which is controlled by xylem
conductivity and reduced by cavitation in the xylem [*Martinez-Vilalta et al.*, 2014;
*Martinez-Vilalta and Garcia-Forner*, 2016]. To avoid leaf desiccation and xylem
cavitation (formulation of air bubbles blocking the ascent of sap flow from the roots to
the leaves) stomatal closure is usually observed during peak daytime sunlight hours in
rainforest canopy species [*Brodribb*, 2003; *Pons and Welschen*, 2003; *Zhang et al.*,
2013]. This reduces the drop in leaf and xylem water potential and thus avoids important
leaf desiccation or xylem cavitation (Figure 12). This type of behavior with strong
stomatal regulation appears to be the norm in the wettest tropical forests [*Fisher et al.*,
2006; *Konings and Gentine*, 2016].
In tall canopy species the flow in the xylem from the roots is limited and cannot
sufficiently rehydrate the upper xylem and leaves, and it cannot be compensated by the
plant internal storage, whereby stomatal shutdown is inevitable to avoid desiccation and
xylem cavitation (Figure 12) [*Phillips et al.*, 1997; 2004; *Lee et al.*, 2005; *Oliveira et al.*,
2005; *Phillips et al.*, 2008; *Scholz et al.*, 2011; *Zeppel et al.*, 2014; *Konings and Gentine*,
2016]. In summary, water stress in tropical rainforest canopy species is not primarily due
to soil water stress but rather to the atmospheric demand and the build up of water stress
in the soil-plant continuum. Radiation, temperature and VPD are therefore essential for
tropical forests further emphasizing the importance of radiation and light on those forests.
Land-surface and ecosystem models, apart from a few exceptions [*Xu et al.*, 2016;
*Kennedy et al.*, 2017], do not represent plant hydraulics and typically only rely on an
empirical reduction of stomatal and ecosystem conductance, and therefore transpiration
and GPP, as functions of root-averaged soil moisture or water potential (e.g., [*Noilhan*
*and Planton*, 1989; *Sellers et al.*, 1996a; 1996b; *Ek*, 2003; *Boulet et al.*, 2007; *Gentine et*





*al.*, 2007; *Ngo-Duc et al.*, 2007; *Stoeckli et al.*, 2008; *Balsamo et al.*, 2009; *Boone et al.*,
2009a; *Bonan et al.*, 2011; *Lawrence et al.*, 2011; *Niu et al.*, 2011; *Bonan et al.*, 2012;
*Canal et al.*, 2014; *Han et al.*, 2014; *Naudts et al.*, 2014; *De Kauwe et al.*, 2015; *Chaney*
*et al.*, 2016; *Chen et al.*, 2016; *Haverd et al.*, 2016] among others). The root profile
averaging of soil moisture or water potential to define water stress exaggerates the impact
of surface drying, as in reality deeper roots may still effectively transport water to the
plant xylem even if surface roots experience dry conditions and therefore can maintain
overall high rates of GPP and transpiration.
The inclusion of plant hydraulics in tall canopy species leads to strong differentiation
between leaf (and upper xylem) and soil water potential (Figure 12) during midday,
especially in the dry season. Indeed, leaf and xylem water potentials substantially drop
because of the large transpiration rates through the stomata and because the xylem cannot
be instantaneously refilled due to the large flow drag in the elongated xylem. As a result,
plant hydraulics induce a shutdown of stomata during the day reducing the transpiration
rate near peak solar hours, also knownknows as "midday depression," in order to reduce
desiccation of the leaf and xylem. In addition, plant hydraulics also induces a natural
hydraulic redistribution of water in the root profile reducing dryness in the upper profile
in the dry season [*Lee et al.*, 2005; *Oliveira et al.*, 2005; *Domec et al.*, 2010; *Prieto and*
*Ryel*, 2014; *Kennedy et al.*, 2017], using deep root moisture rather than surface soil
moisture when needed, as the water flows down gradient of water potentials. This is
fundamentally different from typical parameterizations using average water stress of the
root water profile, which are oversensitive to surface water stress, in typical
parameterizations [*Kennedy et al.*, 2017]. Both of those effects lead to reduced sensitivity
to water stress [*Kennedy et al.*, 2017] and help maintain higher rates of transpiration
throughout the entire dry season [*Kennedy et al.*, 2017], whereas typical land surface
models overestimate water stress in the dry season [*de Gonçalves et al.*, 2013;
*Alemohammad et al.*, 2016; 2017].





## 4   Land-atmosphere interactions – local and nonlocal

### 4.1   Local feedback and heterogeneity – shallow clouds (fog and shallow convection)

We suggest that that the most critical land-atmosphere feedbacks in tropical rainforests involve shallow clouds and fog rather than deep convective clouds. Clearly, much of the focus of tropical land-atmosphere interactions has been on feedbacks involving precipitating deep convection, and the impact of heterogeneity on convective rainfall. On the other hand, the coupling of the land surface to radiation has been relatively understudied. Shallow clouds lead to reduced productivity and transpiration [*Anber et al.*, 2015a], yet the latter depends on cloud thickness as cumulus (shallow convection) generate more diffuse light and can boost photosynthesis when they are not too thick [*Ouwersloot et al.*, 2017] . Fog on the other hand, strongly diminishes the amount of light received by the ecosystems. Fog [*Anber et al.*, 2015a] and shallow clouds [*Giangrande et al.*, 2017] appear to be one of the primary differences between the dry and the wet season (in addition to the preferential occurrence of nighttime mesoscale convective systems in the rainy season, which are not directly relevant for land-atmosphere interactions associated with daytime processes). Low-level cloudiness largely affects the surface incoming radiation by reducing shortwave surface incoming radiation in the wet season, especially in the morning [*Anber et al.*, 2015a; *Giangrande et al.*, 2017], which in turn leads to strong reduction in GPP and ET. These clouds are also tightly connected to surface processes and especially the surface energy partitioning. Indeed nighttime fog, which often persists into the early daylight hours, is largely induced by longwave temperature cooling, especially in the presence of evening rain in the wet season, which generates dew formation [*Anber et al.*, 2015a]. Shallow clouds are themselves directly forced by surface-generated thermals due to boundary layer processes [*de Arellano et al.*, 2014], and they are modified by the sensible and latent heat flux magnitude [*de Arellano et al.*, 2014]. Shallow convection and low-cloud cover are also tightly connected to the seasonality of the forest and to the diurnal cycle [*Anber et al.*, 2015a; *Tang et al.*, 2016; *Giangrande et al.*, 2017].

Historically, the study of land-atmosphere interactions in the Tropics, and tropical rainforests in particular, has emphasized effects of heterogeneity, especially due to



deforestation, on the generation of deep convection through mesoscale circulations (see
[*Lawrence and Vandecar*, 2015] for a complete review, as well as [*Avissar and Pielke*,
1989; *Pielke and Avissar*, 1990; *Pielke et al.*, 1991; *Dalu et al.*, 1996; *Avissar and*
*Schmidt*, 1998; *Taylor et al.*, 2007; 2009; 2011; *Rieck et al.*, 2015; *Khanna et al.*, 2017]).
The hypothesis behind this is that deforestation reduces EF and surface roughness
[*Khanna et al.*, 2017]. The associated increased buoyancy flux over the deforested areas,
mostly reflecting a shift toward increased sensible heating, induces mesoscale
circulations. These circulations enhance cloudiness through local buoyancy fluxes,
turbulent kinetic energy generation, and low-level moisture advection from adjacent
forested areas, thus providing all the key ingredients for moist convection generation
[*Rieck et al.*, 2014; 2015]. It seems unlikely however that momentum roughness plays a
major role in this high radiation environment [*Park et al.*, 2017], where circulations are
mostly buoyancy-driven. Instead, the heat and moisture roughness lengths [*Park et al.*,
2017] as well as leaf area index and stomatal conductance, which scales the magnitude of
the evapotranspiration flux, are the main players, in addition to changes in soil moisture
availability, for the circulation.
Induced mesoscale circulations and associated deep convection are clearly observable
with remote sensing observations [*Khanna et al.*, 2017] and are more important in the dry
season [*Khanna et al.*, 2017], when convection is more locally, and regionally, triggered
[*Anber et al.*, 2015a; *Ghate and Kollias*, 2016]. Once precipitation occurs though, cold
pools, i.e., density currents induced by ice melt and evaporating rain in downdrafts,
dominate the surface-induced mesoscale circulation [*Rieck et al.*, 2015], and reduce the
surface heterogeneity signal. In the wet season, the relative contribution of local forcing
to the total rainfall is small as the bulk of the precipitation is due to mesoscale convective
systems or larger-scale systems propagating throughout the basin, less tightly connected
to surface and boundary layer processes [*Ghate and Kollias*, 2016].
Even during the dry season, a large fraction of the Amazon and of Indonesia only
experience minimal water stress (Figure 9 and Figure 8) so that increased radiation
generates higher rates of photosynthesis (Figure 6) and ET (Figure 7) [*Anber et al.*,
2015a]. As such the radiation feedback of mesoscale-induced clouds may systematically
impact clearings and deforested regions (Figure 13) and are more systematic and longer
lasting than mesoscale-induced convective rainfall. Fewer studies have studied changes





in shallow clouds [*Wang et al.*, 2000; *Lawton et al.*, 2001; *Chagnon et al.*, 2004; *Ray et*
*al.*, 2006; *Wang et al.*, 2009; *Pielke et al.*, 2011; *Rieck et al.*, 2014; *Anber et al.*, 2015a],
even though the impact of changes in the surface energy partitioning and heterogeneity
on low-level clouds is clear and spatially systematic (Figure 1). Given the importance of
cloud cover on shortwave radiation and their importance for the differentiation between
the dry and wet seasons over wet tropical rainforests we believe that this low-cloud
feedback might be quite critical for rainforest ecosystem functioning. Indeed it was
pointed out by [*Morton et al.*, 2014; *Anber et al.*, 2015a; *Morton and Cook*, 2016] that
light changes between the dry and wet season due to changes in cloud cover were one of
the primary reasons for changes in the seasonality of surface fluxes, in addition to leaf
flush out [*Lopes et al.*, 2016; *Saleska et al.*, 2016]. We also note that the shading due to
low clouds reduces surface temperature and ecosystem respiration [*Mahecha et al.*, 2010;
*Peterhansel and Maurino*, 2011; *Thornley*, 2011; *Hadden and Grelle*, 2016; *Ballantyne et*
*al.*, 2017]. So, cloud-induced reductions in respiration can cancel reductions in
photosynthesis, such that the net effect of cloud shading on net ecosystem exchange is
unclear. In an academic study inspired in the thermodynamic characteristics in the
Amazonia, [*Horn et al.*, 2015] showed that coupling with the surface leads to a change in
the length scales that characterized clouds, and a reduction of the cloud life time. As a
result, there are larger populations of smaller shallow cumuli.
In addition to regulating radiative energy balance at the surface, [*Wright et al.*, 2017]
have shown that shallow convection transports moisture, provided by plants'
transpiration, from the atmospheric boundary layer to the lower troposphere during the
late dry season and early dry to wet transition seasons (July-September). This
mechanism, referred to as the "shallow convective moisture pump", plays an important
role in priming the atmosphere for increasing deep convection (e.g., [*Schiro et al.*, 2016]
[*Zhuang et al.*, 2017]), and wet season onset over the Amazon [*Wright et al.*, 2017].
The results discussed until now omitted the relation between physical processes and the
atmospheric composition, and more specifically the role of chemical reactions and
aerosol. Over rainforests, the pristine and undisturbed conditions of the atmospheric
boundary layer described in the seminal study by [*Garstang and Fitzjarrald*, 1999] are
currently undergoing rapid changes due to atmospheric composition modifications. Their
direct impact on the radiative and microphysical properties are due to biomass burning





and enhancement of concentrations of secondary organic aerosol precursors. Biomass
burning in Amazonia leads to increase aerosol optical depth and to abnormal distributions
of the heating rate profile. Analyzing systematic experiments performed by large-eddy
simulations, [*Feingold et al.*, 2005] studied the processes that lead to the suppression of
clouds. Firstly, at the surface there is clear indications that the latent and sensible heat
flux are reduced, yielding convective boundary layers characterized by less turbulent
intensity and by delays in the morning transition [*Barbaro and Arellano*, 2014]. Both
aspects tend to reduce cloud formations. Secondly, [*Barbaro and Arellano*, 2014]
indicated that the vertical location of the smoke layer is crucial in determining how the
cloud characteristics*, i.e* cloud cover, will change. As described by [*Feingold et al.*,
2005], smoke confined in the well-mixed sub-cloud layer might positively benefit the
cloud formation since it distributes the heat uniformly that contributes to enhance
convection. On the other hand, smoke layers located within the cloud layer tend to
stabilize the cloud layer and therefore decrease the possibility of cloud formation. These
results are very much dependent on the aerosol optical properties defined by their
heating, scattering and hygroscopic properties. As a first indicative figure, the mentioned
LES study and observations by [*Koren et al.*, 2004] stressed that smoke layers with an
aerosol optical depth larger than 0.5 might already lead to cloud suppression by 50%. [*Yu
et al.*, 2008] have shown observationally that the influence of aerosols on shallow clouds
varies with meteorological conditions. When the ambient atmosphere is drier (relative
humidity ≤60%), the aerosol induced cloud burning effect (evaporation of cloud droplets)
due to increased absorption of solar radiation by aerosols out-weight the increase of cloud
droplets due to aerosol-cloud microphysical effect. The reduced shallow clouds can
further enhance the surface dryness. In contrast, when the ambient atmosphere is
relatively humid (relative humidity ≥60%), the aerosol-cloud microphysical effect out-
weighs the cloud burning effect, leading to an increase of shallow clouds and relative
humidity near surface. In so doing, aerosols can amplify the original moisture anomalies
near the surface. Aerosols have also shown to increase of the life time of mesoscale
convection over Congo and Amazon, due to delay of the precipitation that enhances ice
formation and increase lifetime of the mature and decay phase of deep convection
[*Chakraborty et al.*, 2016].





These modifications are not only related to the direct emission of aerosol, but also to
changes in the gas phase chemistry that act as a precursor for the formation of secondary
organic aerosol. [*Andreae et al.*, 2002] already described the differences in $NO_x$ and
ozone ($O_3$) mixing ratio depending on the Amazonia site. From rather pristine conditions
with $NO_x$ and ozone levels below 0.1 ppb and 20 ppb, to values above 0.1 ppb and
maximum levels of $O_3$ near 50 ppb.  Recent field experiments within the Green Ocean
Amazon campaign (GoAmazon) (Fuentes et al., 2016; [*Martin et al.*, 2016] corroborate
these levels as well as the high levels of the bio-organic compounds, in particular
isoprene and monoterpene. Closely related, these changes are accentuated by
anthropogenic emissions, i.e. Manaus. The unique distribution of aerosols in Amazonia
might explain observed differences in deep convection, in particular lighting frequency,
between Amazonia, the Maritime continent and the Congo basin [*Williams et al. 2004*].
To represent these chemistry changes and their effect on convection adequately, the
dynamic effect that drive processes such as the entrainment of pollutants from the free
troposphere need to be taken into account [*Vila-Guerau de Arellano et al.*, 2011]. As a
result of this interaction between radiation, the land surface, dynamics and chemical
processes, the transition from turbulent clear convective conditions to shallow cloudy
convection may be modified in the future. Current efforts in monitoring them and
improving the parameterizations of convection are under way [*Dias et al.*, 2014]. These
efforts should include also in an integrated manner the combined role of dynamics and
chemistry to quantify relevant processes like the ventilation of pollutants from the sub-
cloud layer into the cloud layer, i.e. mass flux parameterizations, under representative
Amazon conditions [*Ouwersloot et al.*, 2013].
In addition to affecting cloud microphysics, biomass burning in the tropics significantly
affects the global carbon budget. For example, in September and October of 2015 fires in
the Maritime continent released more terrestrial carbon (11.3 Tg C)  than the
anthropogenic emissions of the EU (8.9 Tg C) [*Huijnen et al.*, 2016]. The extent of forest
fires in this region is tied to El Niño-induced drought conditions, and antecedent SST
patterns are closely related to burned area at the global scale, particularly in hotspots
concentrated in the tropics [*Chen et al., 2016*]. Aerosol emissions and biomass burning
exert a strong control on land-atmosphere coupling of the carbon and water cycles, and
the consequences of this coupling is observable globally.



1.1. Nonlocal feedback – deep convection and large-scale circulation

Thus far, we have largely viewed land-atmosphere coupling through the lens of local
conditions, but how should we modify this view in light of remote influences (see WTG
discussion) or coupling between local and larger-scale conditions? Here we illustrate
some aspects of how land-atmosphere coupling in the Tropics is impacted by the larger-
scale.
4.1.1    Large-scale coupling, idealized modeling
Consider the Lagrangian tendency equation for conservation of atmospheric water
vapor, expressed in terms of specific humidity $q$:
$\frac{dq}{dt} = S(q)$                    (3)
where $S(q)$ is the sum of sources and sinks of specific humidity. In the absence of
sources and sinks, (3) implies that the specific humidity of a parcel of air is conserved
following the atmospheric flow. In what follows, we consider a vertically-integrated
form of (3) such that:
$\langle \frac{\partial q}{\partial t} \rangle = E - P - \langle \mathbf{v_H} \cdot \nabla q \rangle - \langle \omega \frac{\partial q}{\partial p} \rangle$             (4)
Here $E$ and $P$ represent, respectively, the surface evapotranspiration source and the
precipitation sink of water vapor, while $\langle ... \rangle$ represents a mass-weighted vertical
(pressure) integral from the surface (at pressure $p_s$) to the nominal top of the troposphere
(at pressure $p_t$), i.e., $\langle ... \rangle = \int_{pt}^{ps} ... \frac{dp}{g}$. The third and fourth terms on the right-hand side
(RHS) of (4) are horizontal and vertical moisture advection. Equation (4) is normalized
such that $\langle q \rangle$ has units of mm, thus effectively corresponding to column water vapor, and
terms on the right hand side are given in units of mm/day. Equation (4) is often used to
construct a diagnostic budget of precipitation, or in perturbation form, precipitation
anomalies. As a caveat, within the tropics, the dominant large-scale balance in deep
convecting regions is typically between vertical moisture advection (or equivalently in
the vertically-integrated form, moisture convergence) and precipitation, which may limit
the utility of (4) in attributing causality.
Using equation (4) as a starting point, [*Lintner and Neelin*, 2007; 2009] constructed a
framework for estimating *where* spatial transitions between tropical non-precipitating and
precipitating conditions, referred to as convective margins, should occur. By coupling

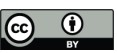



the water and energy (surface and atmosphere) equations, and invoking WTG and
convective quasi equilibrium assumptions, as well as a zero-surface flux constraint over
land, [*Lintner and Neelin*, 2009] derived the following expression for locating the
convective margin, $x_c$, along a prescribed inflow air-mass trajectory from an initial point
over the ocean onto land (see Figure 15 for a schematic overview):
$$x_c = L_c \ln\left[\frac{q_c + q_E}{q_0 + q_E}\right]$$ (5)
$L_c$ denotes a length scale defined as $\frac{v_q Ms}{Mq_p(R_{toa,net} - E)}$ where $v_q$ is the mean horizontal wind
field, weighted with respect to the vertical moisture profile. From the WTG temperature
equation, and subject to the zero net surface flux constraint over land, the divergent
component of the large-scale circulation can be related to the net TOA radiative heating,
$R_{toa,net}$. $Ms$ is the dry static stability and $Mq_p$ the vertical moisture stratification per unit
moisture. The moisture values $q_0$, $q_c$, and $q_E$ denote, respectively, the initial inflow air
mass moisture, a moisture-related threshold for initiation of deep convection,; and a
moisture scale associated with evapotranspiration over the inflow path, $q_E = \frac{v_q E}{L_c}$. Because
of vertical integration, these quantities are column integrated values.
Note that the advantage of coupling the atmospheric moisture equation to the temperature
equation is that under the WTG approximation, the divergent component of the flow is
itself diagnosed, which can be instructive for identifying mechanisms involved. Also, in
equation (5) and the definition of $q_E$, we have assumed that $L_c > 0$ and that convergent
low-level flow is signed positive.
Evapotranspiration gives rise to two opposing effects on $x_c$. First, with increasing $E$ , $q_E$
should increase, which causes $x_c$ to decrease, i.e., moistening from evapotranspiration
experienced along the inflow path leads to the convective margin being reached closer to
the inflow point. Second, as $E$ increases, $L_c$ increases: this can be understood as the
indirect effect of $E$, acting through reduction of convergence along the flow path, which
shifts the onset point for deep convection away from the inflow point (see Figure 15).
Lintner et al. (2013) developed an idealized prototype for diagnosing large-scale land-
atmosphere coupling constructed from the idealized temperature and moisture equations
used in developing the convective margins model described, but further coupled to a
simple bucket soil moisture model. From this model, [*Lintner et al.*, 2013] derived an





analytic expression for the sensitivity of precipitation to soil moisture variation from
which it is possible to infer dependences on key model parameters, such as the timescale
for convective adjustment (assumed in the Betts and Miller-type convection scheme
applied), cloud-radiative feedback strength, and surface turbulent flux exchange.
[*Schaefli et al.*, 2012] developed a conceptually similar model from which an analytic
expression for the ratio of evaporated moisture integrated along flow path to precipitation
(or recycling ratio) was obtained (Figure 16). We suggest that such idealized model
frameworks, which consider tropical land-atmosphere interactions by coupling both water
and energy cycles, should continue to be brought to bear on observations as well as more
sophisticated regional or global climate or earth system models, as they can be helpful in
diagnosing linkages between local and non-local feedbacks.
### 4.1.2   Coupling
[*Green et al.*, 2017] recently developed a method to define the feedback between the
biosphere and atmosphere using multivariate conditional Granger causality (based on
lagged autoregressive vectors). We here use a similar framework using ET from
WECANN and precipitation from GPCP as well as photosynthetically active radiation
from CERES (Figure 18).
Most of the feedback between surface ET and precipitation occurs in the spatial
transitional, Monsoonal regions, such as the Savanna region of Northeastern Brazil, the
Monsoonal region of the Sahel and Southern Africa, as well as India and Northern
Australia. In Brazil, these results are consistent with the above-mentioned concept of
convective margin and the impact of soil moisture and transpiration rate on the location
of the transition between the dry and wet regions. The Sahelian and Southern African
Monsoon are also located in regions between very dry (deserts) and humid regions, where
surface feedback may be crucial for the penetration of the Monsoonal flow inland
[*Lintner and Neelin*, 2009; *Lintner et al.*, 2015]. Indeed, the biosphere in this region
modulates the local climate state: multiple equilibrium states, corresponding to different
ecosystem initial conditions, exist under the same external forcing [*Wang et al.*, 2000].
The effect of vegetation on land-atmosphere coupling manifests itself at multiple
timescales. At short timescales after precipitation, evaporation is accelerated with
intercepted water in the canopy. However, at longer timescales vegetation acts to delay
and prolong evaporation of water stored in the root zone. The magnitude and timescale of





these sources of water recycling will vary depending on ecosystem structure, including
rooting depth and canopy structure, which may co-evolve with atmospheric conditions at
the interannual timescale [*Nicholson*, 2000]. This represents a clear pathway for two-way
feedbacks between the land surface and precipitation.
We further emphasize that those feedbacks (Figure 18) are likely to also be influenced by
non-local conditions, with regional and large-scale changes in ocean to land flow and the
in-land distance of penetration influencing local coupling. We note that climate models
seem to exhibit soil moisture (and therefore evapotranspiration)- precipitation feedbacks
in similar tropical regions, when averaged across models, even though individual model
response varies [*Koster et al.*, 2011; *Seneviratne*, 2013] (one degree pixel and monthly
time scales). We emphasize that the PAR radiation product is very uncertain in the
tropics [*Jim nez et al.*, 2011] as it ultimately relies on a model to obtain surface incoming
radiation, which might explain the reduced feedback strength. It is also likely that the
bulk of the radiative feedbacks are taking place at smaller times scales such as the ones
observed with MODIS (Figure 14). This shallow cloud cover is relatively steady spatially
and in time, especially in the dry season.
### 4.1.3   Moisture tracking and source attribution
A fundamental consideration in the study of the hydrologic cycle over tropical
continents is where the moisture for precipitation ultimately derives. As [*van der Ent et*
*al.*, 2010; *van der Ent and Tuinenburg*, 2017] note, this consideration is not merely of
academic interest: indeed, it is quite likely that anthropogenic modification of the land
surface has altered terrestrial evapotranspiration (as well as runoff) to impact
precipitation. A common approach to moisture source attribution over tropical land
regions involves deriving air mass histories using Lagrangian trajectories. Such
trajectories are obtained by temporally integrating the 3-dimensional wind field to
estimate the positions of idealized air mass parcels through time. Trajectories can be
computed in either a forward or backward sense: the latter are initialized from an arrival
point and integrated backward through time. Combining a Lagrangian back trajectory
approach with rainfall and leaf area index data, [*Spracklen et al.*, 2012] quantified the
linkage between downstream rainfall amount and upstream air mass exposure to
vegetation (Figure 17). Over more than half of the tropical land surface, the Spracklen et
al. estimates indicate a twofold increase in downstream rainfall for those air masses




passing over extensive vegetation compared those passing over little upstream vegetation.
Based on these estimates and extrapolating current Amazonian deforestation trends in the
future, these authors project wet and dry season rainfall decreases of 12 and 21%,
respectively, by the year 2050.
Other analyses using air mass histories have demonstrated the significance of terrestrial $E$
sources for remote land regions. For example, [*Drumond et al.*, 2014] used the
FLEXPART model forced with ERA-Interim reanalysis to estimate $E - P$ along
trajectories passing over the La Plata Basin in subtropical South America to establish that
much of the moisture entering this region derives from the Amazon Basin to the north
and west.

### 4.1.4   Seasonality and seasonal transitions
One of the outstanding issues in the study of tropical land region climates involves
controls on precipitation seasonality, particularly its regional variability. To leading
order, the seasonality follows the variation in maximum solar heating, but other factors,
such as ocean thermal inertia, topography, dynamics and circulation, and moisture
transport, as well as the state of the land surface, can exert considerable influence on the
timing and amplitude of tropical land region seasonal evolution.  Over the Amazon basin,
seasonality exhibits marked variation in both latitude and longitude:  for example, at 5S,
the dry-to-west transition proceeds from the central Amazon eastward toward the Atlantic
coast [*Liebmann and Marengo*, 2001].  It is also worth noting a pervasive tendency for
the dry-to-wet season transition to occur much more rapidly than the wet-to-dry
transition, as evident in tropical monsoon systems including South Asia, West Africa, and
South America.

Analyzing multiple observational and reanalysis products, [*Fu and Li*, 2004]
identified a strong influence of surface turbulent fluxes on the dry-to-wet transition and
its interannual variability over the Amazon.  In particular, their results link earlier wet
season onset to wetter conditions in the antecedent dry season:  the higher latent fluxes at
the end of a wetter dry season encourage weaker convective inhibition (CIN) but
enhanced CAPE, both of which are more favorable to wet season rainfall occurrence.
However, these authors also underscore the participation of the large-scale circulation
and its role in establishing a background environment (e.g., moisture convergence) to





support wet season rainfall.  Incursion of cold fronts into the southern Amazon may act as
triggers for rapid initiation of wet season onset once the local thermodynamics become
favorable [*Li et al.*, 2006].
Recent researches suggest that the land-atmospheric coupling plays a central role in
determining the earlier timing of the wet season onset over western and southern
Amazonia, relative to that of eastern Amazonia. Both in situ and satellite ecological
observations have consistently shown that rainforests increase their photosynthesis, thus
evapotranspiration (ET), during late dry season across Amazonia (e.g., [*Huete et al.*,
2006; *Lopes et al.*, 2016; *Munger et al.*, 2016; *Wehr et al.*, 2016]). The wet season onset
over the southern hemispheric western and southern Amazonia occurs during September
to October, about two to three months before the arrival of the Atlantic ITCZ [*Fu et al.*,
2016]. Using several satellite measurements, including deuterium (HDO) of the
atmospheric water vapor and SIF, Wright et al (2017) have shown that such an increase
of ET in the late dry season is the primary source of increasing water vapor in the lower
troposphere that initiates the increase of deep convection and rainfall over southern
Amazonia. In particular, the increase of water vapor with enriched HDO in the boundary
layer and free troposphere, follows the increase of photosynthesis during late dry season.
The HDO value of the atmospheric moisture is too high to be explained by transport from
Atlantic Ocean, and is consistent with that from plant transpiration.  Such a moistening of
the atmosphere starts in western southern Amazonia, the part of Amazonia that is most
remote from the Atlantic Ocean with high biomass. It then progresses towards eastern
southern Amazonia. Thus, during the late dry season this appears to contribute to the
timing and spatial variation of the initial moistening of the atmosphere, that ultimately
lead to wet season onset over southern Amazonia.
Wet season onset over southern Amazonia has been delaying since the late 1970s
[*Marengo et al.*, 2011; *Fu et al.*, 2013]. In addition to the influence of global circulation
change, such a change has been attributed to land use. For example, [*Butt et al.*, 2011]
have compared long-term rainfall data between deforested and forested areas over part of
the southern Amazonia. They observed a significant delay in wet season onset over the
deforested areas, consistent with that implied by Wright et al. (2017). In addition, [*Zhang
et al.*, 2008; 2009] have shown that biomass burning aerosols, which peak in late dry



season, can also weaken and delay dry to wet season transition by stabilizing the
atmosphere, reducing clouds and rainfall.

## 5   Discussion - conclusions

In this review paper, we have discussed some of the important aspects of land-
atmosphere interactions pertaining to the tropics. This review article is by no means
exhaustive but rather provides insights into some of the important coupled land-
atmosphere processes at play in the tropics and in rainforest ecosystems in particular.
We have argued that feedbacks between the land surface and precipitation in the tropics
are possibly non-local in nature and mostly impact moisture advection from the ocean
and the position of deep convection onset. Local rainfall feedback associated with
mesoscale heterogeneities appear to be rather small in magnitude, at least compared to
the annual-mean rainfall, and not sufficiently spatially systematic to truly affect
ecosystem functioning.
Moreover, we contend that land surface-cloud feedbacks, especially those involving
shallow clouds and fog, are critical in terms of regulating light (direct and diffuse),
temperature, and water vapor deficit over tropical forest, but such feedbacks have
received relatively little attention. Remote sensing platforms provide useful information
for quantifying such feedbacks, but these need to be complemented by ground
measurements (especially of photosynthetic rates and respiration). Eddy-covariance
measurements may prove difficult to use, as mesoscale circulations alter the homogeneity
assumption of eddy-covariance methods.
We have also discussed errors and biases in the representation of tropical continental
climates in current generation climate and Earth system models. The average soil
moisture-precipitation feedback strength across earth system models (based on the
GLACE experiment) [*Koster et al.*, 2004] tend to exhibit land-precipitation feedbacks in
similar transitional regions as the ones observed, which seems to be mostly related to
modification of the moisture advection penetration distance from the ocean rather than to
local feedbacks. These feedbacks appear to be of relatively minor importance in the core
of tropical rainforests but are more critical for more marginal rainfall regions (savanna).
These regions are of critical importance for the terrestrial global carbon cycle, providing
the main terrestrial sink, but might be severely impacted by climate change and droughts





in particular [*Laan Luijkx et al.*, 2015]. Whether the interannual variability in surface $CO_2$
flux in those regions is a zero-sum game with wet years compensating dry years still is an
open question especially in the context of rising $CO_2$ concentration.
The core of rainforests seems to be more affected by radiation feedbacks at relatively
small spatial scales (~1km), which can be dramatically influenced by land cover and land
use change. Projected rates of future deforestation are poorly constrained, especially
regionally, though in recent years, the Congo and Indonesia have experienced increasing
deforestation while the deforestation rate in the Amazon has dropped.
Earth system models tend to predict very diverse responses to global warming leading to
broad spread in the capacity of rainforests to continue to act as net carbon sinks [*Swann et*
*al.*, 2015] in the future. Indeed, in the Amazon in particular, the models' response varies
from becoming much drier to more humid. El Niño events are sometimes thought as a
proxy of global warming in the tropics [*Pradipta et al.*, 2016] as they warm the free-
troposphere. Nonetheless for continents the change in the Walker circulation associated
with El Niño may strongly differ from the change associated with a more uniform sea
surface temperature warming in future climate. In particular, mature El Niño events are
associated with strong subsidence over Indonesia, increased ascent off the coast of Peru
but reduced precipitation over the Amazon basin and a relatively neutral response over
the Congo basin. With SST warming across the tropics, the Maritime continent will most
likely become wetter [*Byrne and O'Gorman*, 2015; *Wills et al.*, 2016]. The fate of the
Amazon basin is less clear, as the climate in the region will be impacted by a
combination of free tropospheric warming stabilizing the atmosphere to deep convection
while warming of the Atlantic enhances the low-level MSE of inflow into the basin.
Additionally, warming-induced changes to large-scale circulation such as the intensity or
orientation of low-level Atlantic trade winds could impact Amazonian precipitation
change. Knowledge of the Congo basin remains limited but it appears that the basin will
become dryer under the combined effect of increased temperature and reduced
precipitation [*Greve et al.*, 2014]. One important question involves how the effect of
rising [$CO_2$] modifies surface energy flux partitioning though changes in stomatal
physiology and modify the regional climate though land-atmosphere interactions
[*Lemordant*, 2016].



*Acknowledgments*. This work was supported by Pierre Gentine's new investigator
grant NNX14AI36G, DOE Early Career grant DE-SC0014203, NSF CAREER and
GoAmazon DE-SC0011094. We would like to acknowledge high-performance computing
support from Yellowstone (ark:/85065/d7wd3xhc) provided by NCAR's Computational
and Information Systems Laboratory, sponsored by the National Science Foundation.



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



Table 1. The surface friction velocity, subcloud layer height (where the minimum of
virtual potential temperature flux occurs), ratio of subcloud layer height and Obukhov
length, ratio of surface friction velocity and Deardorff convective velocity scale, and the
total number of identified clouds for 12 time instants in each case.

| Case | S3 | S2 | S1 | CTL | R1 | R2 | R3 |
|---|---|---|---|---|---|---|---|
| $u_* u_*$ [m s$^{-1}$] | 0.07 | 0.14 | 0.21 | 0.28 | 0.35 | 0.42 | 0.56 |
| $z_i$ [m] | 590 | 590 | 590 | 590 | 590 | 610 | 630 |
| $z_i / L$ | 392.1 | 49.0 | 14.5 | 6.1 | 3.1 | 1.9 | 0.8 |
| $u_*/ w_*$ $u_*/ w_*$ | 0.10 | 0.20 | 0.30 | 0.40 | 0.50 | 0.60 | 0.79 |
| $N_{cloud}$ | 2248 | 2229 | 2283 | 2302 | 2250 | 2703 | 2776 |














List of Figures





Figure 1: Snapshot of cloud cover over the Amazon basin (courtesy NASA, MODIS
visible bands) in the dry season. Small clouds are shallow convective clouds, highlighting
surface Bowen ratio changes between the river and the forest. At the bottom right, the
deep convective cells, does not follow the surface heterogeneity (and is much larger in
scale).Figure 1: Snapshot of cloud cover over the Amazon basin (courtesy NASA,
MODIS visible bands) in the dry season. Small clouds are shallow convective clouds,
highlighting surface Bowen ratio changes between the river and the forest. At the bottom
right, the deep convective cells, does not follow the surface heterogeneity (and is much
larger in scale).
Figure 2: Diurnal cycle in local hour of dry (red) and wet (blue) season observations of
precipitation at K34, near Manaus, along with their standard deviation averaged across
years 2010-2014.
Figure 3: Response of tropically-averaged free tropospheric temperature between 700mb
and 200mb to El Niño Southern Oscillation (choosing the ENSO 3.4 index)
Figure 4: Seasonal variations in Evapotranspiration (ET) from WECANN, Precipitation
(Precip) based on GPCP, Net Radiation (Rn) from CERES and Gross Primary Production
(GPP) based on WECANN informed by Solar-Induced Fluorescence (SIF) over the wet
part of the Amazon (top left), the Savanna region of Brazil (top right), over Indonesia
(bottom left) and over the Congo basin (bottom right).
Figure 5: Seasonality of Precipitation based on GPCP in the tropics in December-
January-February (a), March-April-May (b), June-July-August (c), and September-
October-November (SON) and its latitudinal average (e).
Figure 6: same as Figure 5 but for Gross Primary Production (GPP)
Figure 7: same as Figure 5 for latent heat flux LE
Figure 8: same as Figure 5 for sensible heat flux H
Figure 9: same as Figure 5 for evaporative fraction (EF), the ratio of LE to H+LE.
Figure 10: same as Figure 5 for sea-level surface moist static energy flux, the sum of
sensible heat flux H and latent heat flux
Figure 11: Schematic showing the vertical structure of light and water limitations in a
tropical forest.



Figure 12: Climatology of the diurnal cycle of leaf water potential and top soil water
potential in the dry and wet seasons in Caxiuana, Brazil simulated by the Community
Land Model (CLM) with plant hydraulics.
Figure 13: Mesoscale heterogeneity impact on cloud generation. a) Typical perspective
regarding the impact of deforestation and clearings generating deep convective clouds
and b) more realistic impact, in terms of mostly a modification of shallow convection
cloud cover, impacting radiation more than precipitation.
*Figure 15*: (a) Schematic of the key elements of the convective margins framework as
applied along an inflow path across northeastern South America. The solid blue and
black lines are precipitation and vertically-integrated moisture, while dashed blue line
corresponds to precipitation smeared out by transients. Adapted from Figure 2 of Lintner
and Neelin (2009). (b) Rainfall longitudinal transects from the Climate Anomaly
Monitoring System (CAMS) raingauge-derived precipitation data for September-
October-November for the period 1950-2000 for El Niño (red), La Niña (blue), and all
(black) years, averaged over 3.75°S-1.75°S. From Figure 4b of Lintner and Neelin

(2007).

Figure 16: adapted from Schäfli et al. (2012)
Figure 18: Land-atmosphere feedback strength (change in the variance due to the
feedback) between Precipitation and ET (top) and Photosynthetically Active Radiation
(PAR) (bottom) based on recent metric developed by Green et al. [2017] using a
multivariate Granger causality approach.



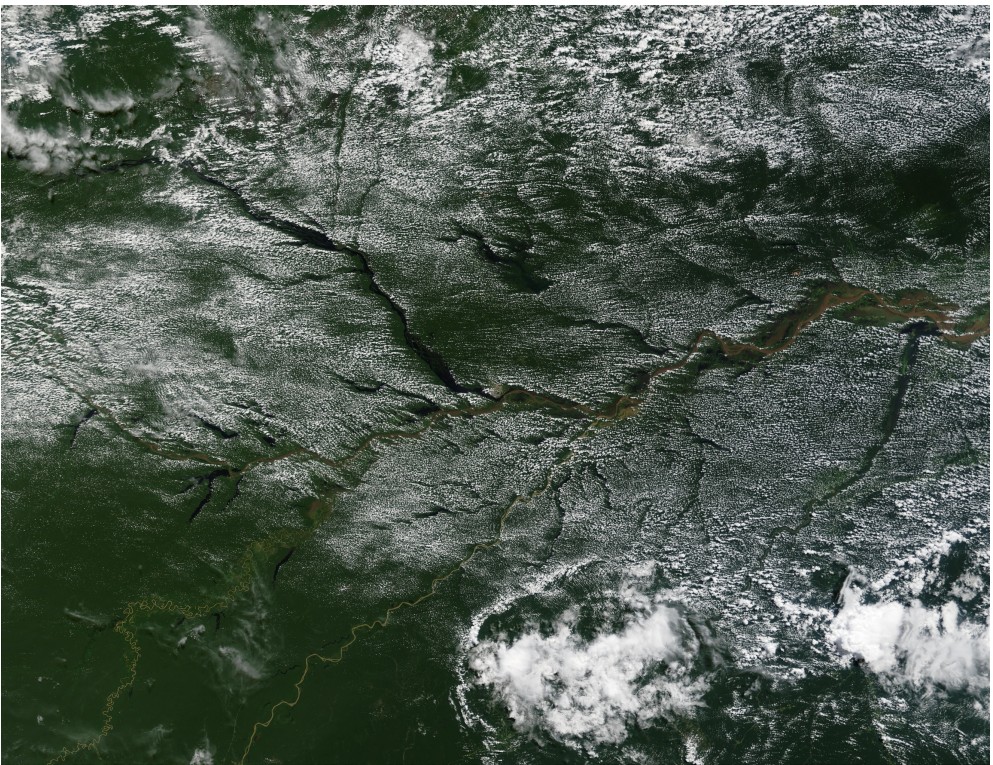


Figure 1: Snapshot of cloud cover over the Amazon basin (courtesy NASA, MODIS visible
bands) in the dry season. Small clouds are shallow convective clouds, highlighting surface Bowen
ratio changes between the river and the forest. At the bottom right, the deep convective cells, does
not follow the surface heterogeneity (and is much larger in scale).





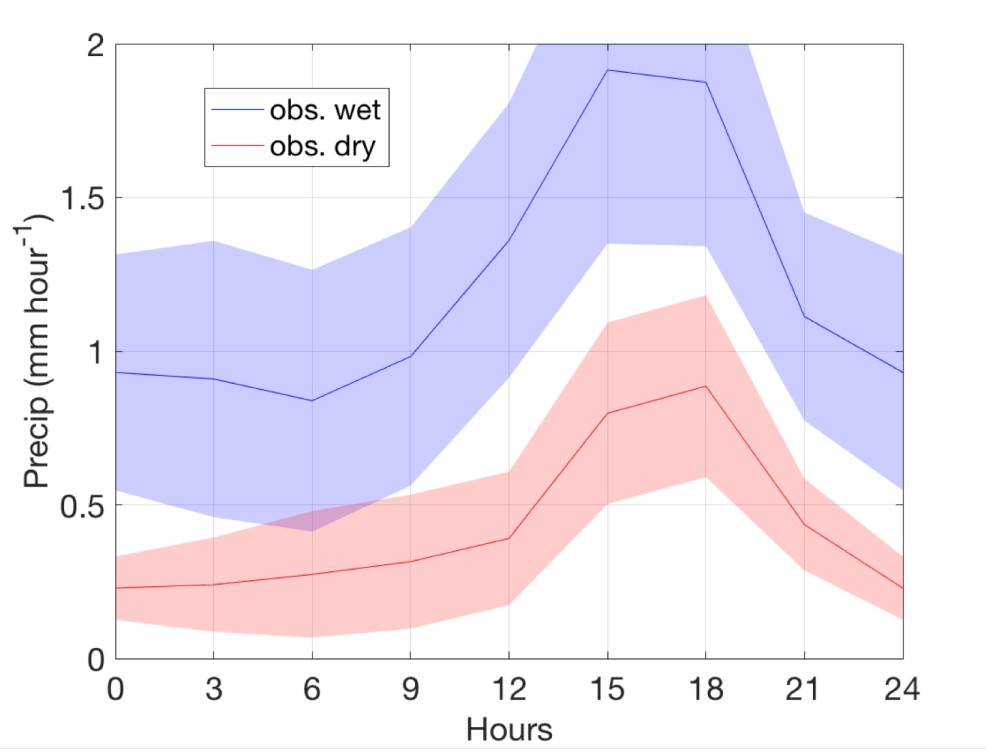

Figure 2: Diurnal cycle in local hour of dry (red) and wet (blue) season observations of precipitation at K34, near Manaus, along with their standard deviation averaged across years 2010-2014.



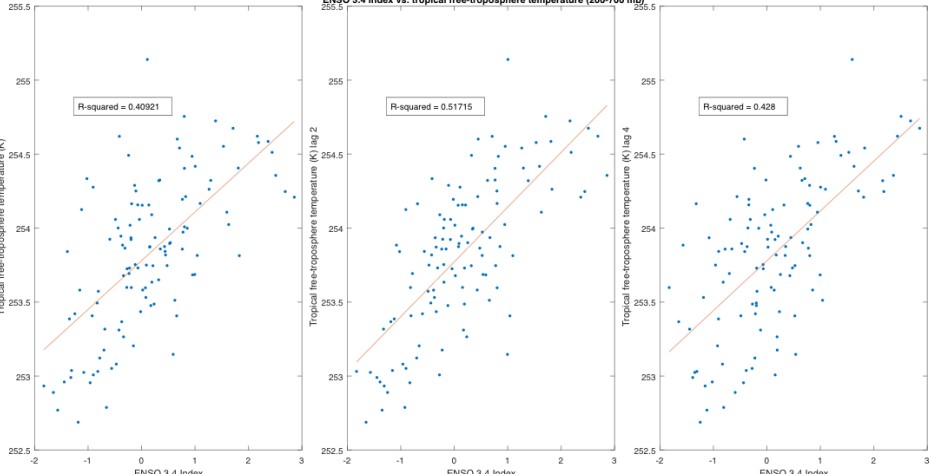


Figure 3: Response of tropically-averaged free troposphere temperature between 700mb and
200mb to El Niño Southern Oscillation (choosing the ENSO 3.4 index) with either no lag (left) or
2-month lag (middle) or 4-month lag (right)

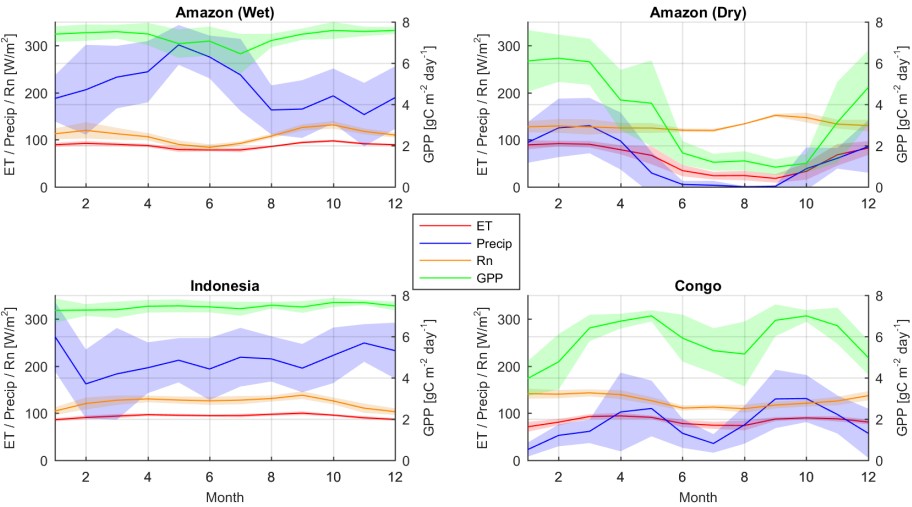


Figure 4: Seasonal variations in Evapotranspiration (ET) from WECANN, Precipitation
(Precip) based on GPCP, Net Radiation (Rn) from CERES and Gross Primary Production (GPP)
based on WECANN informed by Solar-Induced Fluorescence (SIF) over the wet part of the
Amazon (top left), the Savanna region of Brazil (top right), over Indonesia (bottom left) and over

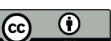



the Congo basin (bottom right).


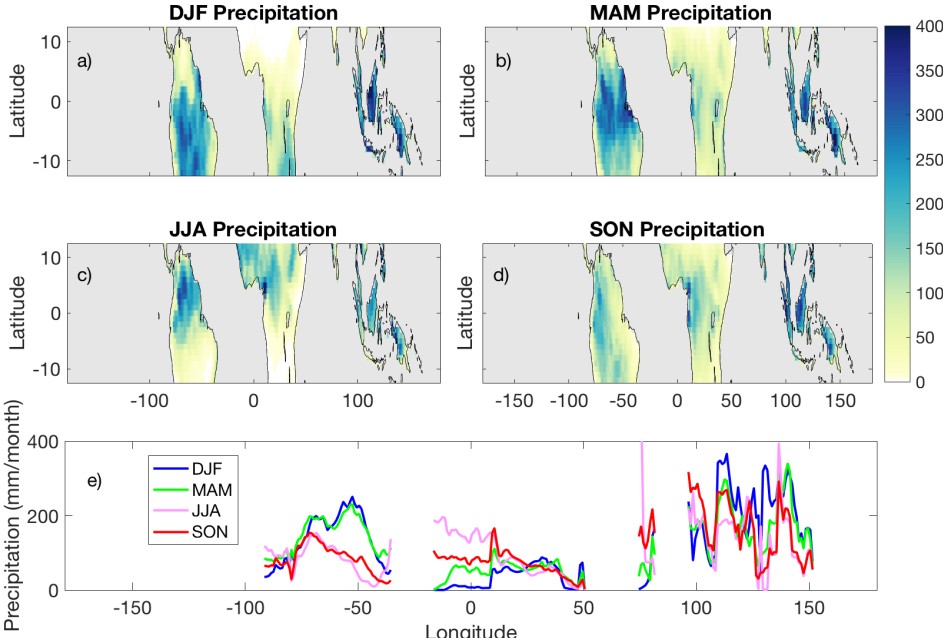


Figure 5: Seasonality of Precipitation based on GPCP in the tropics in December-
January-February (a), March-April-May (b), June-July-August (c), and September-
October-November (SON) and its latitudinal average (e).




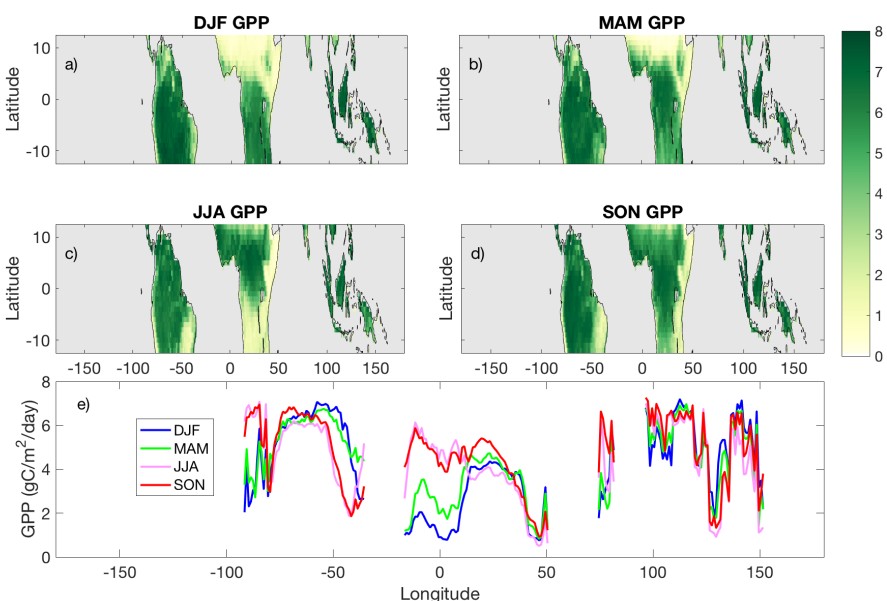


Figure 6: same as Figure 5 but for Gross Primary Production (GPP)




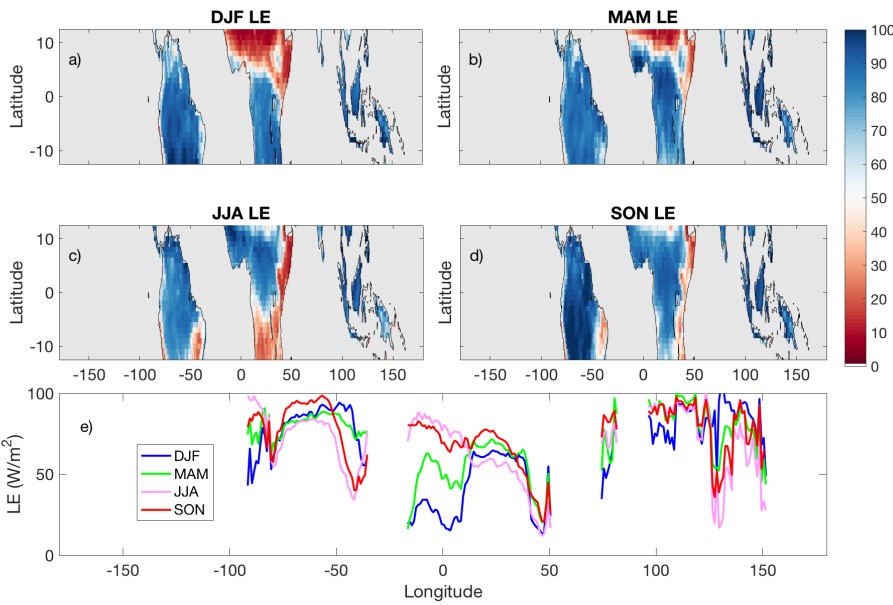


Figure 7: same as Figure 5 for latent heat flux LE

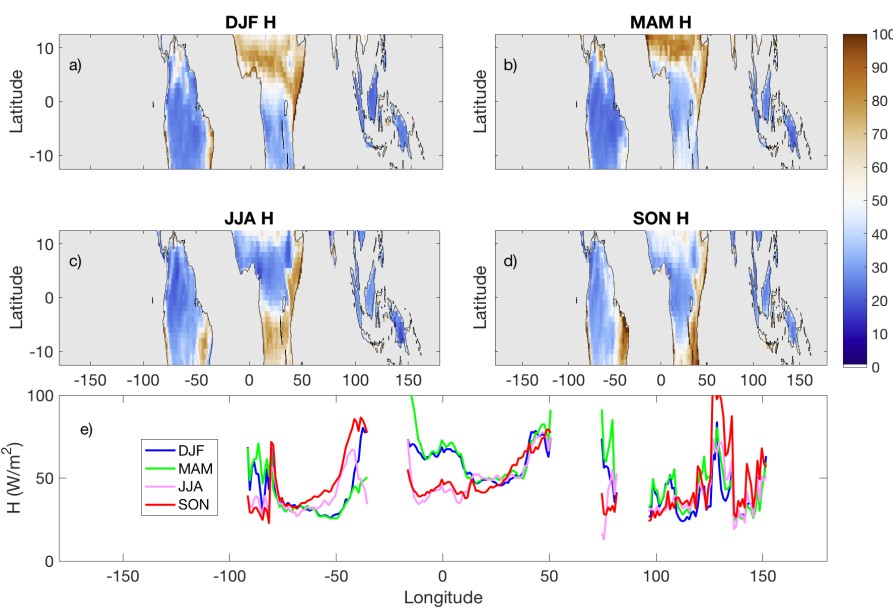




Figure 8: same as Figure 5 for sensible heat flux H



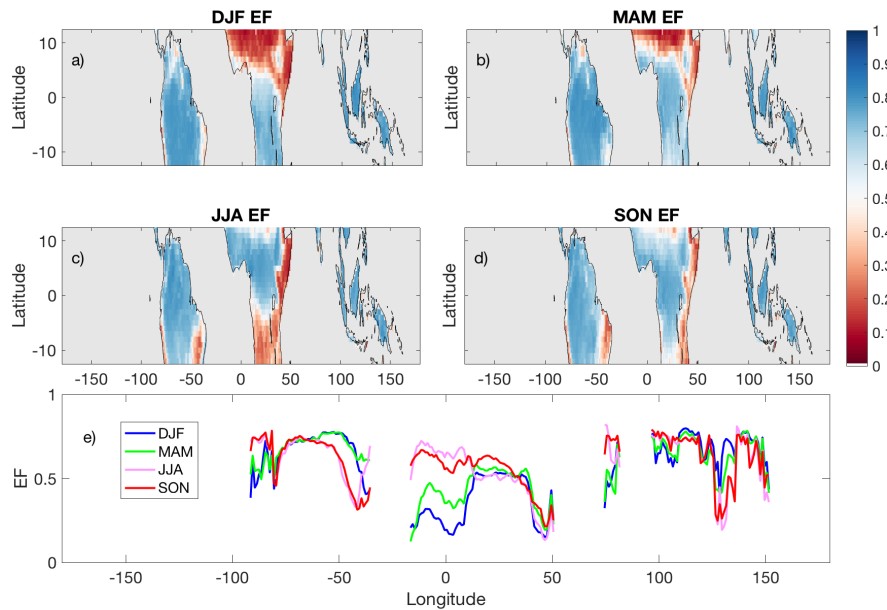


Figure 9: same as Figure 5 for evaporative fraction (EF), the ratio of LE to H+LE.









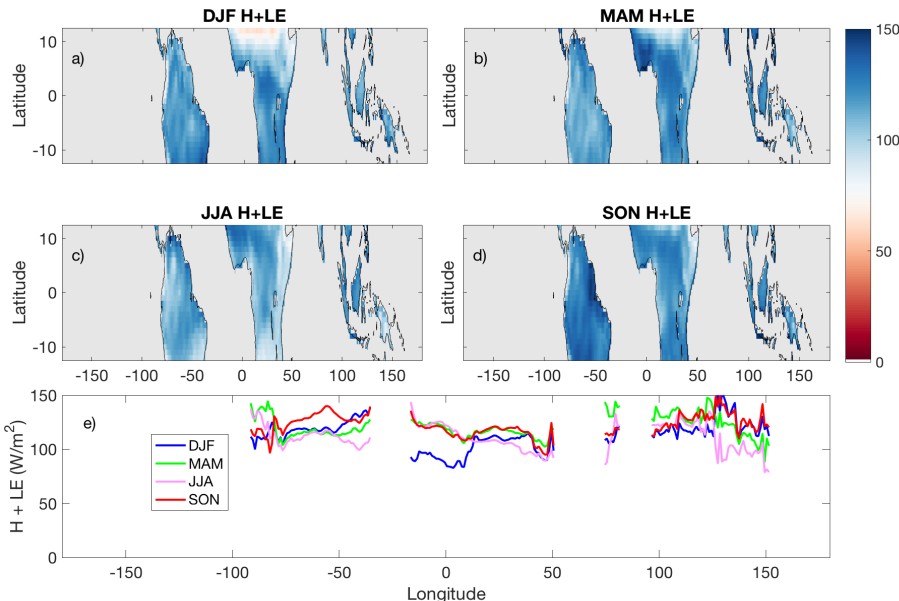


Figure 10: same as Figure 5 for sea-level surface moist static energy flux, the sum of sensible

heat flux H and latent heat flux





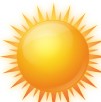

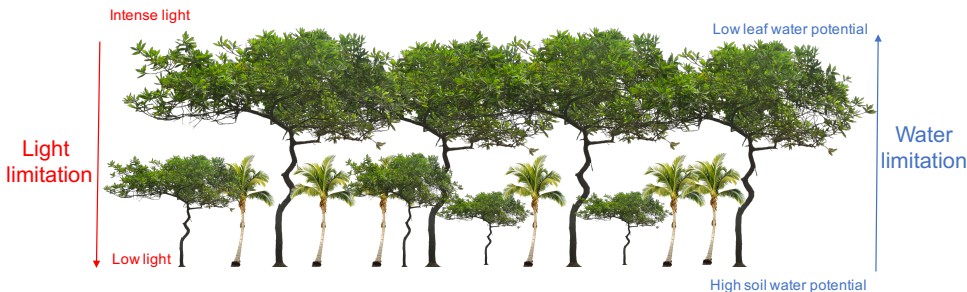




Figure 11: Schematic showing the vertical structure of light and water limitations in a tropical
forest.


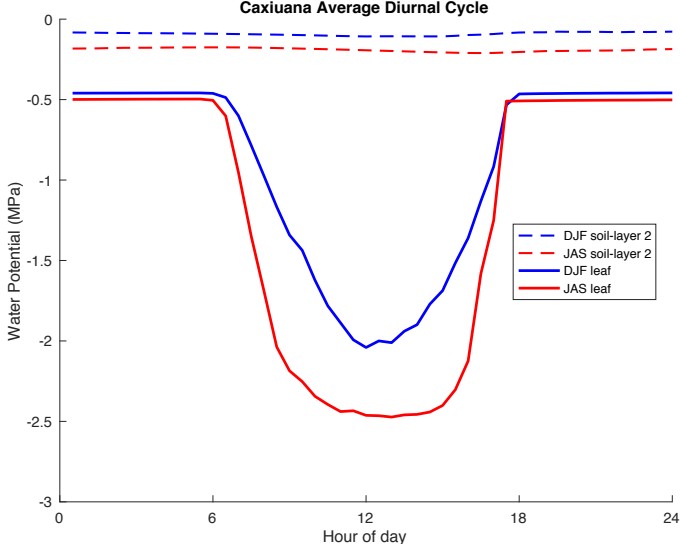


Figure 12: Climatology of the diurnal cycle of leaf water potential and top soil water
potential in the dry and wet seasons in Caxiuana, Brazil simulated by the Community
Land Model (CLM) with plant hydraulics.

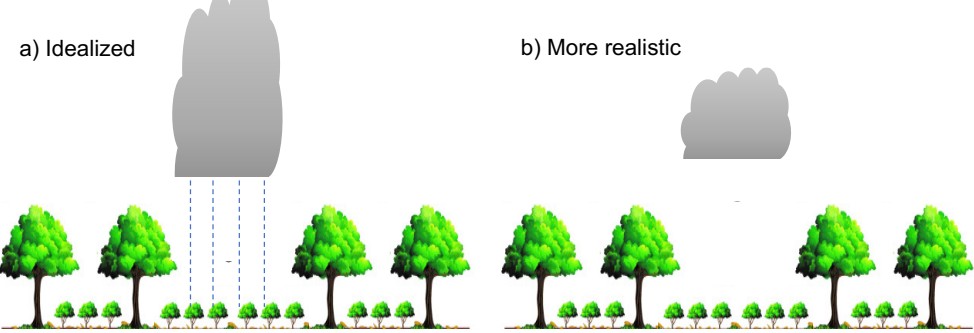


Figure 13: Mesoscale heterogeneity impact on cloud generation. a) Typical perspective
regarding the impact of deforestation and clearings generating deep convective clouds and b)
more realistic impact, in terms of mostly a modification of shallow convection cloud cover,
impacting radiation more than precipitation.





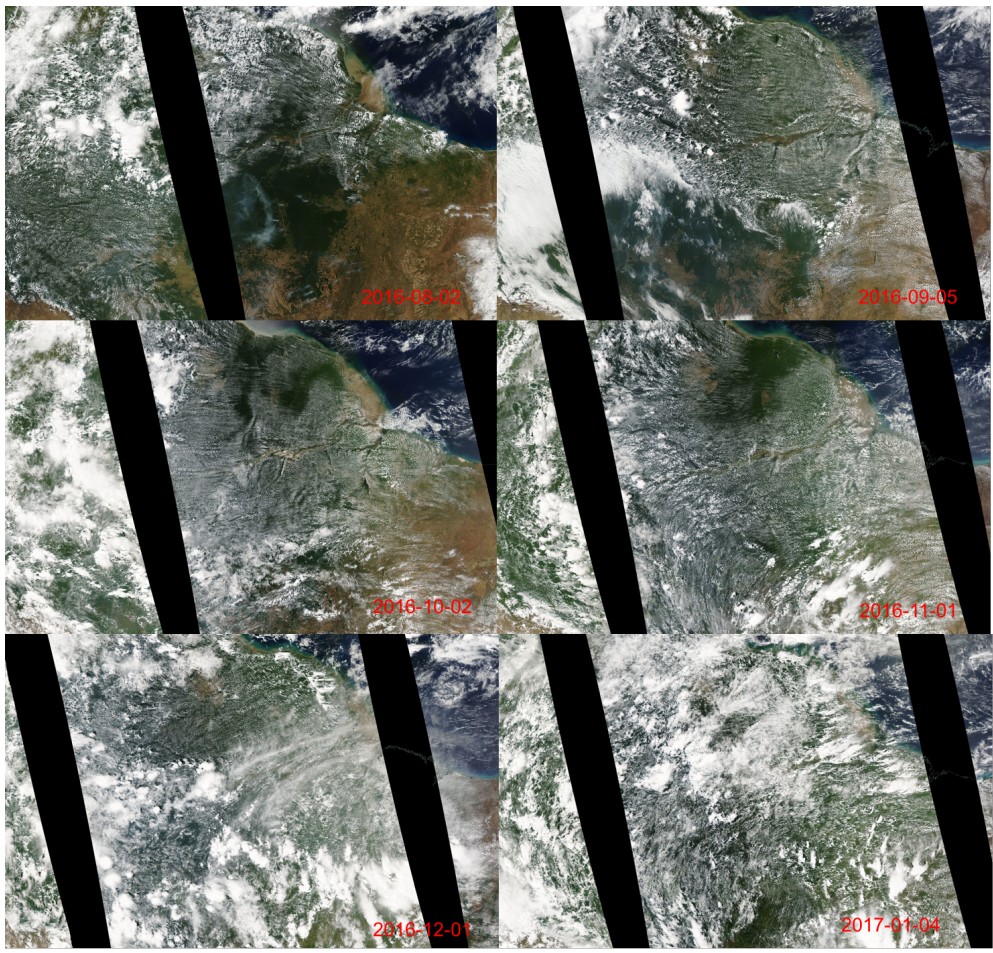


Figure 14: MODIS visible image of the Northwestern Amazon as the basin transition into
the wet season. In the dry season surface heterogeneity whether due to rivers, forest-
deforested patches or land-ocean contrast are very clear. In the wet season those sharp
gradients disappear as cloud cover mostly dominated by deep convection starts
organizing at scales independent from the surface heterogeneity.





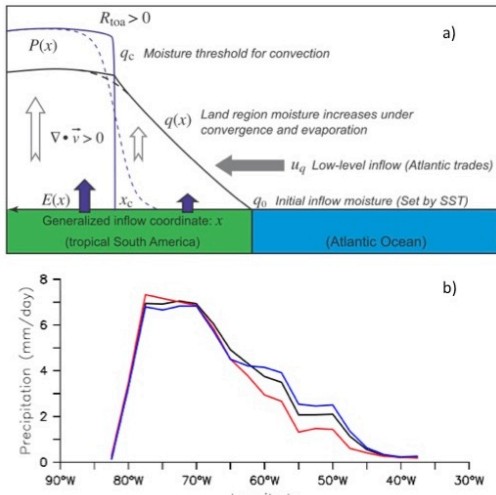


Figure 15: (a) Schematic of the key elements of the convective margins framework as applied along an inflow path across northeastern South America. The solid blue and black lines are precipitation and vertically-integrated moisture, while dashed blue line corresponds to precipitation smeared out by transients. Adapted from Figure 2 of Lintner and Neelin (2009). (b) Rainfall longitudinal transects from the Climate Anomaly Monitoring System (CAMS) raingauge-derived precipitation data for September-October-November for the period 1950-2000 for El Niño (red), La Niña (blue), and all (black) years, averaged over 3.75°S-1.75°S. From Figure 4b of Lintner and Neelin (2007).

1884

1885

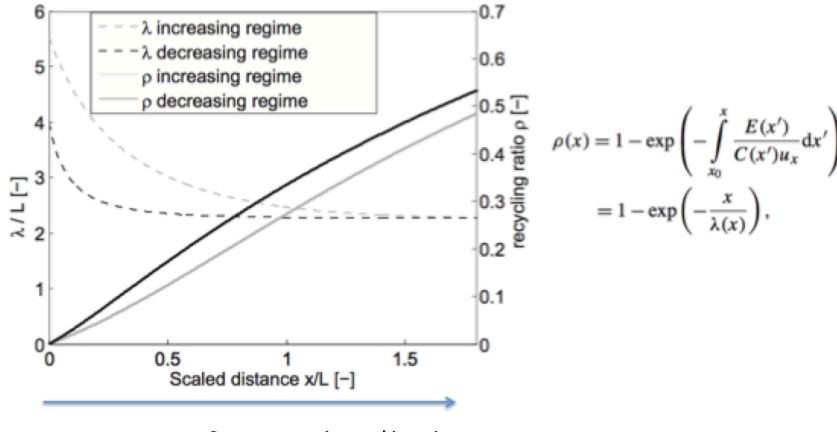

1886

Figure 16: adapted from Schäfli et al. (2012)



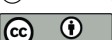

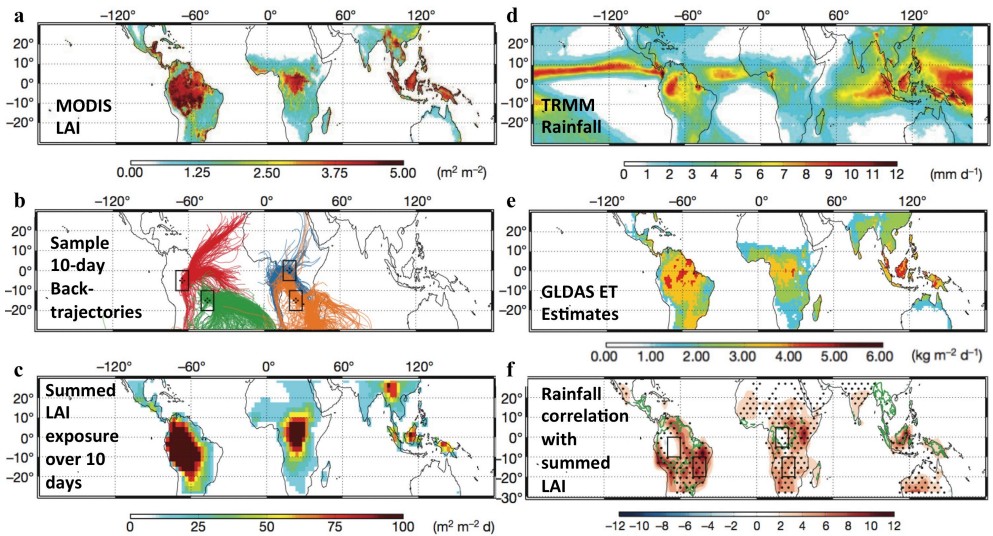

Figure 17: 10 day-backtrajectory analysis over several continental regions of the continental tropics, along with LAI, mean TRMM estimated rainfall, and GLDAS ET estimates.

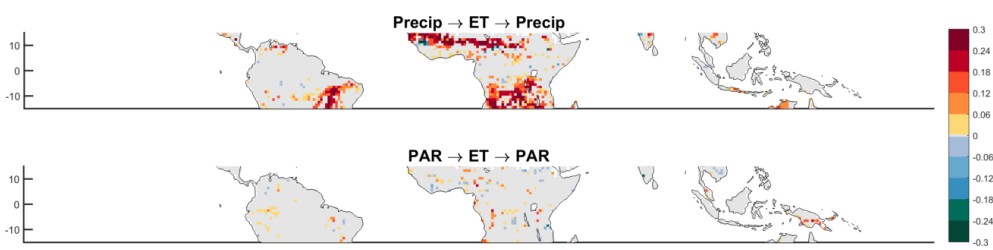

Figure 18: Land-atmosphere feedback strength (change in the variance due to the feedback) between Precipitation and ET (top) and Photosynthetically Active Radiation (PAR) (bottom) based on recent metric developed by Green et al. [2017] using a multivariate Granger causality approach.