# Peer review of "Land-atmosphere interactions in the tropics – a review 2 3 Pierre Gentine Department of Earth and Environmental Engineering, 4 5 Earth Institute 6 Columbia University, New York, NY, USA 7 8 Adam Massmann 9 Department of Earth and Environmental Engineering, 10 Earth Institute 11 Columbia Unive"

_Hydrology and Earth System Sciences, 2019_

## Referee Comment (RC1) · Anonymous Referee #1 · 17 Mar 2019

General comments:

The authors present a nice review of a topic that has not been properly reviewed before - the specific range of land-atmosphere interactions in the tropics. Most published work including reviews have focused on the subtropics and summertime mid-latitudes. This is a welcome addition to the literature. The topic will draw a diverse readership with different interests and expertise.

The authors need to be sure they are not jumping in at too technical a level or assuming too much foreknowledge of the readers. Also, the paper could use more homogenization in style and level of detail - it is clear that different authors wrote different parts. It needs to be made more even throughout.

In Sec 3.2 and later in Sec 5 the problem that LSMs have representing water stress and GPP should explicitly mention root access to groundwater / shallow water tables

in tropical lowlands and plains - a process that is not present in most models with their very shallow soils. This is hinted at, but there should be explicit statement regarding the link between tropical phenology and hydrology.

Specific comments:

L133: A comma after "initiation"

L195-196: The equivalence drawn between turbulent carbon fluxes and GPP needs a little explaining for non-expert readers.

L242: The work of Tawfik and colleagues (10.1002/2013GL057984, 10.1175/JHM-D-14-0117.1, 10.1175/JHM-D-14-0118.1) is highly germane here as well.

Figure 1: Please give the date and time of the image (a la Fig 14), and the domain (lon and lat range) - also a scale superimposed on the figure would be helpful to understand the size of the clouds.

L250-259: I suggest this paragraph be reordered, grouping the density/buoyancy processes (thermals, radiative destabilization, cold pools) and references first, and then the dynamically forced vertical motions (circulations, wave activity).

L260-268: Propagating convection should also be mentioned here - I am thinking of work by Nieto-Ferreira and Rickenbach, for instance.

L277: A glitch in the citation software for the Lintner reference.

Figure 3 is not referenced anywhere - please remove.

L326: "Section XX".

L338: Remove reference to Figures 4-8 - they do not correspond exactly to what is said in this sentence, and they get referenced properly later.

Figure 4: Need to explain that it is lambda*P that is shown, not P.

Figures 5-10: These can be made easier to read. Please confine the zonal range to

95°W-165°E, stack the maps in 4 rows, instead of 2x2, so they are not so distorted, include the units prominently in caption, not just hidden in the Y axis label of panel e. Also, this paper clearly defines Tropics as within 15° latitude of the equator, but these maps stop around 12° - why?

L374: This is misleading - the plot appears to have a steady annual cycle of precipitation because of the meridional averaging. There is in fact locally distinct seasonalities in precipitation in most locations - a point that is mentioned later. Please note here the effect of the meridional averaging.

L405: I find the reference to "moist static energy flux" here and in Fig 10 to be clumsy. I know what the authors are trying to say. To my mind, MSE naturally includes the "gz" term. So this statement seems to include orographic forcing (upslope flows) which I believe is not the intention here as this is all locally calculated neglecting horizontal motions over terrain.

L428: This should be Fig 11, not Fig 3.

L430-432: As phrased, this is not a sentence.

Discussion of Fig 12: It is a nice figure, but it is not clear that moisture stress is essentially VPD - this can be clarified in the text. In fact, the whole discussion (L434 onward) is a bit circular and muddled - it could be said much more clearly and more succinctly.

L479-480: The first reference should be to Dawson (1993; 10.1007/BF00317442) - his work was seminal.

L484-486: Don't need to cite the same paper for each phrase. There are several places where the same references are repeated unnecessarily, sentence by sentence.

L502: Change "one" to "two".

Figure 13 would benefit from a schematic diagram that illustrates the contrasts between wet and dry seasons.

[Figure]

L567: Change "inspired in" to "inspired by".

L580: "aerosol" -> "aerosols"

L588: Change "is clear" to "are clear".

L605: "out-weight" -> "outweigh"

L624: Change "i.e. Manaus" to "e.g., from Manaus".

L625: "lighting" -> "lightning"

L647: Section number should be 4.1.

Figure 14: Again, a scale would be a helpful addition.

Sec 4.1.1: This is a complete departure in style from the tone of the rest of the paper. The rest of the paper provides synopses and literature reviews on the various topics, but this is a singular specific conceptual model presented in detail and in vacuo. Spracken et al. (2012; 10.1038/nature11390), Makarieva et al (2013; 10.1175/JHM-D-12-0190.1) and Gimeno (2014; 10.1002/2014WR015477) come to mind as relevant publications on this topic that could provide context - there are certainly others. But this section needs to be made to fit better with the rest of the manuscript.

Figure 15: The blue line looks about the same color as the black line - needs to be more distinct. And what is the dashed black line that cuts the corner around about x_c?

Sec 4.1.3: There is other relevant literature that informs this topic, e.g., Dirmeyer et al. (10.1175/JHM557.1; 10.1016/j.jhydrol.2008.11.016; 10.1175/JHM-D-13-053.1), Keys et al. (2012; 10.5194/bg-9-733-2012), Hoyos et al. (2018; 10.1007/s00382-017-3653-6), to name a few. This section is relatively weak and terse compared to others - there is more than can be said for such a review.

Figure 17 cited out of order, after Fig 18 - and the second occurrence of the word

"continental" can be removed from the caption.

L790: "west" -> "wet"

L806: "researches" -> "research"

L873-: This is new material not discussed earlier - in fact, this mention of climate change responses seems tacked on as a means to exit the paper. It should probably be covered in the core of the manuscript if it is to be mentioned here.

―――――――――――――――――

---

## Referee Comment (RC2) · Anonymous Referee #2 · 1 Apr 2019

Review of "Land-Atmosphere Interactions in the tropics"

The authors present new perspectives based on recent literature, emphasizing the role of surface radiation in biosphere-atmosphere interactions and the water cycle. This is a much needed shift in focus toward (shallower) clouds and aerosols and their coupling to the surface water balance. A central part of this coupling, and a focus of the review, is on transpiration and its connection to clouds and aerosols via surface radiation and photosynthesis. These are important yet often overlooked topics for a wide range of current research problems from Earth system modeling to monitoring changes in the water cycle. The review gives a balanced discussion of observations, theory, and modeling, including new techniques to constrain the photosynthesis-water cycle connection from observations. Thus I believe this review will be a valuable contribution.

There are some relatively minor edits and clarifications needed, along with a few sug-

gestions below.

Some of section 4 on WTG approximation could be put into a broader context as a way to study multiscale interactions by parameterizing the larger scales. The discussion of the literature on WTG certainly raises awareness of the challenges in linking the larger and smaller scales and provides a way to gain understanding. The motivation for thinking about nonlocal coupling could be clarified slightly, since I don't think the authors are arguing for nonlocal coupling as being dominant over or even separate from the other. For feedbacks, it seems less clear a priori which scales should be most important for future change in the water cycle; and from a model development perspective, the unknown still centers largely on local or subgrid scale processes (e.g. the diurnal cycle of clouds), although the interactions of the 'nonlocal' and 'local' processes are certainly part of that unknown. It seems the challenge is to make progress on modeling the multiscale and multicomponent system, and in gaining some understanding (and capability of observing) the overall behaviors of the complex system related to water cycle extremes. This sort of discussion would help wrap up the review in the conclusions.

Technical comments/clarifications:

133 their initiation [and] the role of surface [processes]

201 – A little more background on the WECANN product would be helpful, specifically what other observations it uses besides SIF in deriving the surface fluxes (if any).

203 plausible interannual [variability]

231 'The distinction between shallow and deep convection remains elusive' - Elusive may not be the right word; perhaps 'imprecise', 'subjective', or 'contextual' would work better

277 Fix citation: {Lintner:2017gm}. 299 One key concept in tropical climate is the Weak Temperature Gradient (WTG) - This could be set up a bit more with another sentence or two, depending on page limits. 310 - "In addition it is relatively straightforward" - necessary? 326 as discussed in Section XX. 335 'upscale to larger scale.' - Redundant?

"In what follows, we evaluate climatologies of evapotranspiration" - Maybe give some idea of what the reader should expect to learn?

Fig. 4 "over the wet part of the Amazon (top left), the Savanna region of Brazil (top right), " - Some readers may confuse the titles Amazon (wet) and Amazon (dry) for wet and dry seasons as opposed to regions. Perhaps replace with Amazon (rainforest) / Amazon (Savanna) or Amazon (rainforest) / Amazon (Cerrado), and then in the caption make the connection between those regions and wet vs dry climates.

347 – Perhaps be more specific to ET components here "canopy evaporation (of inter-cepted rain)"

Fig. 5 Could you set the aspect ratio of the panels to make this less stretched out and easier to read? i.e., make the axes labels consistent and crop the ocean regions. I recommend stacking all the panels vertically so that panel 3 has the same longitude axis as the seasonal plots. That way we get a clear picture of how variable the SE Asian/Indonesian region is due to the topography (as noted in the text).

Fig. 5 could be better integrated with the text - perhaps add references around line 367 'the topography and the distribution of island land masses leads to strong local variability [Fig. 5e]'

392-395 – "The seasonal pattern of ET resembles GPP..." - this section could use minor editing by breaking up the sentences and expanding to be more specific and clear.

389 "regions (Figure 5). GPP is maximized during the wet season in South America, as GPP is" - The text moves on to GPP without much transition here; perhaps add a transition sentence.

404 It would help to define "moist static energy flux" as LH+SH 419 I assume references for 'why do most contemporary land-surface models incorrectly represent the wettest rainforest GPP and ET...'' are in the prior sections? If so it may help to add a link here

to refer readers back to the introduction.

422 You may mean that capturing this accurately will require better understanding? 430 remove "because" in "because relative humidity is high..." 454 "build up of water stress in the soil-plant continuum" - it may help to introduce the water potential terminology a bit earlier here, since it appears in the next paragraph anyway. That terminology may help to clarify this sentence.

476 Fix "also known as..." 476 Regarding midday depression, there are some references on this for tropical forests (Malhi et al., 1998; Williams et al., 1998; Harris et al., 2004).

492 "We suggest that the most critical land-atmosphere feedbacks..." - It would help to specify this a bit, as to whether it is critical for understanding, addressing ESM water cycle deficiencies, modeling dynamic vegetation in a changing climate, etc...

510 longwave cooling? 512 generates dew or forms dew

548 - "As such the radiation feedback... may systematically impact clearings and deforested regions". I suggest expanding and editing this sentence to reflect the three ideas it contains. The first is that transpiration is able to buffer the dry season effects in these regions, stabilizing ET, so that the feedback loop involving precipitation and ET is weakened. Thus, the impact of the dry season on ET (and hence clouds) is strongest in mesoscale clearings and deforested regions. In addition, the feedback of shallower clouds and surface radiation may be more important than the feedback of deeper clouds and precipitation.

564 - Here and in a few other places there is some discussion on respiration, in which the link to the water cycle may be lost for some readers as it is not as clear as for photosynthesis. Consider clarifying those connections.

605 "outweigh" 613 "increased lifetime" 618 "depending on the Amazonia site, from rather pristine..." ? 628 dynamics that drive

631 "the transition from turbulent clear convective conditions to shallow cloudy maybe modified in the future" - Do you mean changes in the frequency of the transition or nature of the transition?

640 – the discussion on Maritime continent biomass burning is nice for geographical balance and is an outcome of precipitation deficits tied to El Nino. The carbon cycle impact of the burning is discussed in the review, but do you think it also has impacts on the water cycle that could be discussed here?

Figure 16 caption – could use a little more information on what increasing/decreasing regime mean

751 'reduced feedback strength' - reduced relative to what?

References Malhi Y., A. D. Nobre, J. Grace, B. Kruijt, M. G. P. Pereira, A. Culf, S. Scott, Carbon dioxide transfer over a Central Amazonian rain forest. Journal of Geophysical Research-Atmospheres 103, 31593-31612 (1998); published online EpubDec 27 (10.1029/98jd02647).

Harris P. P., C. Huntingford, P. M. Cox, J. H. C. Gash, Y. Malhi, Effect of soil moisture on canopy conductance of Amazonian rainforest. Agricultural and Forest Meteorology 122, 215-227 (2004); published online EpubApr 20 (10.1016/j.agrformet.2003.09.006).

Williams M., Y. Malhi, A. D. Nobre, E. B. Rastetter, J. Grace, M. G. P. Pereira, Seasonal variation in net carbon exchange and evapotranspiration in a Brazilian rain forest: a modelling analysis. Plant Cell and Environment 21, 953-968 (1998); published online EpubOct (10.1046/j.1365-3040.1998.00339.x).

---

## Referee Comment (RC3) · Anonymous Referee #3 · 22 Apr 2019

As a review paper, there are not really any new findings here, but rather a summary of previous work. The paper does a fine job of this, and I believe it will be a valuable resource for others. I recommend acceptance with minor revisions. That being said, I do have some comments.

Lines 202-203: When I looked at the WECANN papers I did not see proof that seasonal cycles were reproduced. I see r-squared values in the tables, but not evidence showing seasonal cycles (also: I think the sentence should have 'variability' added at the end). I'm not sure I'm convinced that WECANN is better than other models in the tropics (really, Brazil. Do we have enough tower data in Africa or the Maritime Continent to really make an assessment?). Is there a way to establish this?

This brings up something else: By using the words 'wet tropics', the implication is that tropical forests do not experience any water stress. I don't agree with this. There are a couple of papers [da Rocha et al., 2009, Costa et al., 2010] that discuss the variation in 'environmental control' (light limitation) and 'biotic control' (water limitation across precipitation and vegetation gradients in Brazil. Yes, the cerrado (savanna) is generally water-limited, and the wettest forest is light-limited, but the transition is not binary, nor is it limited to the cerradao (transition forest). I think a discussion of light- and water-limitation across gradients, and our uncertainty about the relative importance of each is an important part of tropical land-atmosphere interaction that is missing from this manuscript. George Vourlitis and coauthors have done some good work in the cerradao that should be described (I'm not going to list them all here). Baker et al. (2013) put forth a conceptual description of this gradient in a modeling study. Whether describing Brazil, the forest-to-desert transitions in Africa (both north and south), or the ecotone in Australia, there is important ecophysiological information in these transitions (and their response to changing climate) that is ignored here.

On a potentially related note, I'm curious if the European Centre Amazon conversion papers need to be mentioned (Cox, Huntingford, Jones, et al.). I understand a review paper is not the same thing as a history paper, but these HADGCM papers got a lot of attention, and actually initiated quite a bit of investigation. The fact that there has been somewhat of a retreat from the initial findings means that these papers are no longer the 'state of the science', but they were seminal, along with the Saleska 2003 paper (in my opinion) in the initiation of some pretty important lines of investigation.

**Figures:**

The figures don't flow smoothly with the text, in that the sequence of figures doesn't match when they are referenced. I was jumping back and forth in the figures as they

were mentioned in the text, and I think a little rearranging would make the readability better.

Labels are too small in Figure 3, and are very difficult to read.

Figures 5-10. The deformation of continents makes these plots hard to interpret. I would prefer to see the horizontal scale of the 4 seasonal plots stretched to match the longitude panel on the bottom of the plot. It would make the plots a little bigger, but readability would be improved. Also, the reader would be able to look directly up from the bottom panel and see the spatial variability in the latitudinal averages.

Figure 16 seems to be thrown in, without much explanation in the text. I'm not sure I understand what is going on here, please clarify.

**Miscellaneous comments:**

Lines 74-75: Could include Friedlingstein (2006) and ?? here as citations.

Lines 84-85: is it worth mentioning previous land-atm coupling papers, like Koster et al. (2004), or Dirmeyer (2011)?

Lines 130-133: Cumbersome sentence, maybe some typos. Please reword.

Line 169: If you say 15 S, you probably don't need the minus sign.

Line 277: Looks like a laTex citation typo.

Line 287: Grabowski (1999) could be cited too.

Line 289: 'models'

Line 324: I think the citation '[*Anber et al., 2015a*]' should be 'Anber et al. [2015a]', since the author name is part of the sentence. This looks like a place where \citep is used instead of \citet in the creation of the manuscript. There are a lot of instances of this in the manuscript, especially from line 587 on.

Lines 398-400: 'Tropically-averaged EF does not evolve much…' With time? With space?

Lines 400-402: incomplete sentence.

Line 406: 'through' not 'though'

Line 428: The text is talking about vertical gradients of light and water availability, yet refers to figure 3, which shows temperature response to ENSO. This recalls the earlier comment about figures and how they are referred to in the text.

Line 430: delete 'because'

Line 476: knownknows

Line 539: delete 'though,'

Lines 549-550: I'm not sure I agree. Mesoscale-Induced clouds may be *initiated* preferentially in clearings and deforested regions, but they don't necessarily *stay* there. When I look at the GOES-16 images over Amazonia, I see clouds moving, not standing still. Do you have evidence that demonstrates that, integrated over time, the cleared/deforested regions are effected by clouds more than non-deforested regions?  This may require some clarification.

Lines 567-569: confusing sentence, please reword.

Lines 574-577: You might want to cite Fu and Li (2006) here as well.

Line 585: 'increased'

Line 625: 'lightning'

Lines 617-620: cumbersome sentences, some rewording would be helpful.

Line 703: inconsistent reference style.

Line 790: 'dry-to-wet'

Lines 815, 832: inconsistent reference style.

Line 893: 'through'

**References**

Baker et al., 2013: Surface ecophysiological behavior across vegetation and moisture gradients in tropical south America. Agricultural and Forest Meteorology, 182-182, 177-188, http://dx.doi.org/10.1016/j.agrformet.2012.11.015

Costa et al., 2010: Atmospheric versus vegetation controls of Amazonian tropical rain forest evapotranspiration: Are the wet and seasonally dry rain forests any different? JOURNAL OF GEOPHYSICAL RESEARCH, VOL. 115, G04021,

    doi:10.1029/2009JG001179, 2010

Da Rocha 2009: Patterns of water and heat flux across a biome gradient from
    tropical forest to savanna in Brazil. JOURNAL OF GEOPHYSICAL RESEARCH, VOL.
    114, G00B12, doi:10.1029/2007JG000640, 2009

Dirmeyer, P, 2011: The terrestrial segment of soil moisture-climate coupling. GRL,
    VOL. 38, L16702, doi:10.1029/2011GL048268, 2011, L16702

Fu, R. and Li, W., 2004: The influence of the land surface on the transition from dry
    to wet season in Amazonia.  Theoretical and Applied Climatology, 78(1-3), 97-
    110.

Koster, R.D. et al., 2004: Regions of strong coupling between soil moisture and
    precipitation. Science, 305, 1138, doi: 10.1126/science.1100217

Rodenbeck C., et al., 2003: $CO_2$ flux history 1982–2001 inferred from atmospheric
    data using a global inversion of atmospheric transport. Atmos. Chem. Phys., 3,
    1919–1964, 2003 www.atmos-chem-phys.org/acp/3/1919/

Vourlitis, G., et al., 2001: Seasonal variations in the net ecosystem CO exchange of a
    mature Amazonian transitional tropical forest (cerradão). *Functional Ecology*
    2001 15, 388–395

Vourlitis, G., et al., 2002: Seasonal variations in the evapotranspiration of a
    transitional tropical forest of Mato Grosso, Brazil. WATER RESOURCES
    RESEARCH, VOL. 38, NO. 6, 1094, 10.1029/2000WR000122

Vourlitis et al., 2008: Energy balance and canopy conductance of a tropical semi-
    deciduous forest of the southern Amazon Basin. WATER RESOURCES
    RESEARCH, VOL. 44, W03412, doi:10.1029/2006WR005526

Vourlitis, G., et al., 2004: EFFECTS OF METEOROLOGICAL VARIATIONS ON THE $CO_2$
    EXCHANGE OF A BRAZILIAN TRANSITIONAL TROPICAL FOREST. *Ecological
    Applications,* 14(4) Supplement, 2004, pp. S89–S100

Vourlitis et al., 2005: The Sensitivity of Diel $CO_2$ and $H_2O$ Vapor Exchange of a
    Tropical Transitional Forest to Seasonal Variation in Meteorology and Water
    Availability. *Earth Interactions* • Volume 9 (2005) • Paper No. 27

---

## Author Comment (AC1) · 14 May 2019

We thank the reviewer for his/her comments and positive assessment of the manuscript.

Comment: The authors need to be sure they are not jumping in at too technical a level or assuming too much foreknowledge of the readers. Also, the paper could use more homogeniza-tion in style and level of detail

Reply: in the revised text we will streamline the discussion and remove unnecessary details

Comment: n Sec 3.2 and later in Sec 5 the problem that LSMs have representing water stressand GPP should explicitly mention root access to groundwater / shallow water tables in tropical lowlands and plains - a process that is not present in most models

with theirvery shallow soils. This is hinted at, but there should be explicit statement regardingthe link between tropical phenology and hydrology. Reply: this is an important comment and indeed we will further emphasize both the varying water table especially near the floodplain but also access to shallow water tables citing recent references.

---

## Author Comment (AC2) · 14 May 2019

We thank the reviewer for his/her positive comments. Specific comment: Some of section 4 on WTG approximation could be put into a broader context as a wayto study multiscale interactions by parameterizing the larger scales. The discussion ofthe literature on WTG certainly raises awareness of the challenges in linking the largerand smaller scales and provides a way to gain understanding. The motivation for think-ing about nonlocal coupling could be clarified slightly, since I don't think the authors arear-guing for nonlocal coupling as being dominant over or even separate from the other.For feedbacks, it seems less clear a priori which scales should be most important forfuture change in the water cycle; and from a model development perspective, the un-known still centers largely on local or subgrid scale processes (e.g. the diurnal cycle ofclouds), although the interactions of the 'nonlocal' and 'local' processes are certainlypart of that unknown. It seems the challenge is to make progress on modeling the multiscale and

multicomponent system, and in gaining some understanding (and ca-pability of observing) the overall behaviors of the complex system related to water cyclee xtremes. This sort of discussion would help wrap up the review in the conclusions. Reply: Indeed the reviewer is correct and we will extend the discussion on scales and how the coupling between scales could be important for convection and extremes.

———————————————

---

## Author Comment (AC3) · 14 May 2019

We thank the reviewer for his/her positive comments.

Review:Land-Atmosphere interactions in the tropics, by Gentine et al.As a review paper, there are not really any new findings here, but rather a summary of previous work. The paper does a fine job of this, and I believe it will be a valuable resource for others. I recommend acceptance with minor revisions. That being said, I do have some comments. Comments: indeed as the reviewer rightly mentioned we aimed at reviewing the state of the art in the field.

Comment: Lines 202-203: When I looked at the WECANN papers I did not see proof that seasonal cycles were reproduced. I see r-squared values in the tables, but not evidence showing seasonal cycles (also: I think the sentence should have 'variability' added at the end). I'm not sure I'm convinced that WECANN is better than other models

in the tropics (really, Brazil. Do we have enough tower data in Africa or the Maritime Continent to really make an assessment?). Is there a way to establish this? Reply: this is a good comment of the reviewer. We will now use additional products in addition to WECANN.

Comment: This brings up something else: By using the words 'wet tropics', the implication is that tropical forests do not experience any water stress. I don't agree with this. There are a couple of papers [da Rocha et al., 2009, Costa et al., 2010] that discuss the variation in 'environmental control' (light limitation) and 'biotic control' (water limitation across precipitation and vegetation gradients in Brazil. Yes, the cerrado (savanna) is generally water-limited, and the wettest forest is light-limited, but the transition is not binary, nor is it limited to the cerradao (transition forest). I think a discussion of light-and water-limitation across gradients, and our uncertainty about the relative importance of each is an important part of tropical land-atmosphere interaction that is missing from this manuscript. George Vourlitis and coauthors have done some good work in the cerradao that should be described (I'm not going to list them all here). Baker et al. (2013) put forth a conceptual description of this gradient in a modeling study. Whether describing Brazil, the forest-to-desert transitions in Africa (both north and south), or the ecotone in Australia, there is important ecophysiological information in these transitions (and their response to changing climate) that is ignored here. Response: we completely agree with the reviewer and it must have been an improper discussion but indeed tropical forests are not binary in their response nor are they uniformly responding (e.g. forest age, biodiversity, height... are key to explain the response of different parts of the Amazon forest). We will correct this.

Comment: On a potentially related note, I'm curious if the European Centre Amazon conversion papers need to be mentioned (Cox, Huntingford, Jones, et al.). I understand a review paper is not the same thing as a history paper, but these HADGCM papers got a lot of attention, and actually initiated quite a bit of investigation.The fact that there has been somewhat of a retreat from the initial findings means that these

papers are no longer the 'state of the science', but they were seminal, along with the Saleska 2003 paper (in my opinion) in the initiation of some pretty important lines of investigation. Response: this is indeed a good point and we will highlight the fact that previous modeling efforts might have overestimated stress and incorrectly represented the feedback with the atmosphere

———————————————————

---

## Author Response (AR1)

General comments:

The authors present a nice review of a topic that has not been properly reviewed before - the specific range of land-atmosphere interactions in the tropics. Most published work including reviews have focused on the subtropics and summertime mid-latitudes. This is a welcome addition to the literature. The topic will draw a diverse readership with different interests and expertise.

The authors need to be sure they are not jumping in at too technical a level or assuming too much foreknowledge of the readers. Also, the paper could use more homogenization in style and level of detail - it is clear that different authors wrote different parts. It needs to be made more even throughout.

Response: we thank the reviewer for the comment: we have now tried to homogenize the different paragraphs so that it would flow better.

In Sec 3.2 and later in Sec 5 the problem that LSMs have representing water stress and GPP should explicitly mention root access to groundwater / shallow water tables in tropical lowlands and plains - a process that is not present in most models with their very shallow soils. This is hinted at, but there should be explicit statement regarding the link between tropical phenology and hydrology.

Response: Indeed, this is a very good comment, we now include a more complete discussion of this issue, in particular citing Maxwell and Condon 2016

Specific comments:

L133: A comma after "initiation"

Response: corrected

L195-196: The equivalence drawn between turbulent carbon fluxes and GPP needs a little explaining for non-expert readers.

Response: we have replaced this with carbon fluxes to avoid any confusion

L242: The work of Tawfik and colleagues (10.1002/2013GL057984, 10.1175/JHM-D- 14-0117.1, 10.1175/JHM-D-14-0118.1) is highly germane here as well.

Response: indeed, this is correct, and has been added (Tawfik et al., 2014, 2015a,b)

Figure 1: Please give the date and time of the image (a la Fig 14), and the domain (lon and lat range) - also a scale superimposed on the figure would be helpful to understand the size of the clouds.

Response: we could not find the same figure but extracted a similar figure from the earth observatory including the scale.

L250-259: I suggest this paragraph be reordered, grouping the density/buoyancy pro- cesses (thermals, radiative destabilization, cold pools) and references first, and then the dynamically forced vertical motions (circulations, wave activity).

Response: This is a good point of the reviewer. This paragraph has been modified accordingly. We have included cold pools in both thermodynamical and dynamical processes

L260-268: Propagating convection should also be mentioned here - I am thinking of work by Nieto-Ferreira and Rickenbach, for instance.

Response: this reference has now been added

L277: A glitch in the citation software for the Lintner reference.

Response: thank you this has now been corrected.

Figure 3 is not referenced anywhere - please remove.

Response: we now reference this Figure when discussing the weak temperature gradient

L326: "Section XX".

Response: there was an issue in the referencing, we have now replaced this by section 4.2

L338: Remove reference to Figures 4-8 - they do not correspond exactly to what is said in this sentence, and they get referenced properly later.

Response: these references have been removed

Figure 4: Need to explain that it is lambda*P that is shown, not P.

Response: indeed this is a good point: we now say: in units of energy (W/m$^2$, by multiplying it by the latent heat of vaporization)

Figures 5-10: These can be made easier to read. Please confine the zonal range to 95°W-165°E, stack the maps in 4 rows, instead of 2x2, so they are not so distorted, include the units prominently in caption, not just hidden in the Y axis label of panel e. Also, this paper clearly defines Tropics as within 15° latitude of the equator, but these maps stop around 12° - why?

Response: We have now modified the maps so they do not look as distorted

L374: This is misleading - the plot appears to have a steady annual cycle of precipitation because of the meridional averaging. There is in fact locally distinct seasonalities in precipitation in most locations - a point that is mentioned later. Please note here the effect of the meridional averaging.

Response: This is an excellent point, indeed we now add the fact that those are regional averages.

L405: I find the reference to "moist static energy flux" here and in Fig 10 to be clumsy. I know what the authors are trying to say. To my mind, MSE naturally includes the "gz" term. So this statement seems to include orographic forcing (upslope flows) which I believe is not the intention here as this is all locally calculated neglecting horizontal motions over terrain.

Response: Here we are discussing the flux and not the state so that there should be no change due to gz when computed at a given level.

L428: This should be Fig 11, not Fig 3.

Response: corrected

L430-432: As phrased, this is not a sentence.

Response: this has been corrected to: "However, water is typically not limiting for low-canopy species, as relative humidity is high and VPD is low, leading to low stress on understory conductance"

Discussion of Fig 12: It is a nice figure, but it is not clear that moisture stress is essen- tially VPD - this can be clarified in the text. In fact, the whole discussion (L434 onward) is a bit circular and muddled - it could be said much more clearly and more succinctly.

Response: in fact, the moisture stress should not just be VPD but rather plant water status/water potential which reflects the interaction of hydraulics (typically not too limiting for tropical rainforest species as recently demonstrated by Liu et al., 2019) and VPD. This is now clarified in the text

Liu, H., Gleason, S. M., Hao, G., Hua, L., He, P., Goldstein, G., & Ye, Q. (2019). Hydraulic traits are coordinated with maximum plant height at the global scale. *Science Advances*, *5*(2), eaav1332. https://doi.org/10.1126/sciadv.aav1332

L479-480: The first reference should be to Dawson (1993; 10.1007/BF00317442) - his work was seminal.

Response: indeed, this is completely correct and has been added now

L484-486: Don't need to cite the same paper for each phrase. There are several places where the same references are repeated unnecessarily, sentence by sentence.

Response: Indeed, we have removed the reference to Kennedy et al, 2017

L502: Change "one" to "two".

Response: corrected

Figure 13 would benefit from a schematic diagram that illustrates the contrasts between wet and dry seasons.

Response: this is a good point from the reviewer. We now emphasize that this schematic was in the dry season. Wet season is more complicated (e.g. mesoscale convective systems).

L567: Change "inspired in" to "inspired by". L580: "aerosol" -> "aerosols"
L588: Change "is clear" to "are clear". L605: "out-weight" -> "outweigh"
L624: Change "i.e. Manaus" to "e.g., from Manaus". L625: "lighting" -> "lightning"

Response: those have all been corrected.

L647: Section number should be 4.1.

Response: there was an issue with the referencing. This has now been corrected.

Figure 14: Again, a scale would be a helpful addition.

Response: this has been corrected

Sec 4.1.1: This is a complete departure in style from the tone of the rest of the pa- per. The rest of the paper provides synopses and literature reviews on the various topics, but this is a singular specific conceptual model presented in detail and in vacuo. Spracken et al. (2012; 10.1038/nature11390), Makarieva et al (2013; 10.1175/JHM- D-12-0190.1) and Gimeno (2014; 10.1002/2014WR015477) come to mind as relevant publications on this topic that could provide context - there are certainly others. But this section needs to be made to fit better with the rest of the manuscript.

Response: this section has been rewritten to better fit the rest of the paper.

Figure 15: The blue line looks about the same color as the black line - needs to be more distinct. And what is the dashed black line that cuts the corner around about $x_c$?

Response: this has been corrected

Sec 4.1.3: There is other relevant literature that informs this topic, e.g., Dirmeyer et al. (10.1175/JHM557.1; 10.1016/j.jhydrol.2008.11.016; 10.1175/JHM-D-13-053.1), Keys et al. (2012; 10.5194/bg-9-733-2012), Hoyos et al. (2018; 10.1007/s00382-017-3653- 6), to name a few. This section is relatively weak and terse compared to others - there is more than can be said for such a review.

Response: the reviewer is correct and this has been corrected and adjusted.

Figure 17 cited out of order, after Fig 18 - and the second occurrence of the word "continental" can be removed from the caption. L790: "west" -> "wet"

Response: those have been corrected

L806: "researches" -> "research"

Response: corrected

L873-: This is new material not discussed earlier - in fact, this mention of climate change responses seems tacked on as a means to exit the paper. It should probably be covered in the core of the manuscript if it is to be mentioned here.

Response: the conclusion has been rewritten
The authors present new perspectives based on recent literature, emphasizing the role of surface radiation in biosphere-atmosphere interactions and the water cycle. This is a much needed shift in focus toward (shallower) clouds and aerosols and their coupling to the surface water balance. A central part of this coupling, and a focus of the review, is on transpiration and its connection to clouds and aerosols via surface radiation and photosynthesis. These are important yet often overlooked topics for a wide range of current research problems from Earth system modeling to monitoring changes in the water cycle. The review gives a balanced discussion of observations, theory, and mod- eling, including new techniques to constrain the photosynthesis-water cycle connection from observations. Thus I believe this review will be a valuable contribution.

Response: we thank the reviewer for the positive assessment of our manuscript.

There are some relatively minor edits and clarifications needed, along with a few sug gestions below.

Some of section 4 on WTG approximation could be put into a broader context as a way to study multiscale interactions by parameterizing the larger scales. The discussion of the literature on WTG certainly raises awareness of the challenges in linking the larger and smaller scales and provides a way to gain understanding. The motivation for think- ing about nonlocal coupling could be clarified slightly, since I don't think the authors are arguing for nonlocal coupling as being dominant over or even separate from the other. For feedbacks, it seems less clear a priori which scales should be most important for future change in the water cycle; and from a model development perspective, the un- known still centers largely on local or subgrid scale processes (e.g. the diurnal cycle of clouds), although the interactions of the 'nonlocal' and 'local' processes are certainly part of that unknown. It seems the challenge is to make progress on modeling the multiscale and multicomponent system, and in gaining some understanding (and ca- pability of observing) the overall behaviors of the complex system related to water cycle extremes. This sort of discussion would help wrap up the review in the conclusions.

Response: this is a good point from the reviewer. We now correct this in the conclusion and expand the discussion regarding key challenges.

Technical comments/clarifications:

their initiation [and] the role of surface [processes]

Response: a comma was missing

– A little more background on the WECANN product would be helpful, specifically what other observations it uses besides SIF in deriving the surface fluxes (if any).

Response: this has been corrected.

plausible interannual [variability]

Response: this has been corrected

'The distinction between shallow and deep convection remains elusive' - Elusive may not be the right word; perhaps 'imprecise', 'subjective', or 'contextual' would work better

Response: we replaced elusive by imprecise

Fix citation: {Lintner:2017gm}.

Response: this has been corrected

One key concept in tropical climate is the Weak Temperature Gradient (WTG) - This could be set up a bit more with another sentence or two, depending on page limits.

Response: we have better connected those sentences and the paragraph.

- "In addition it is relatively straightforward" - nec- essary?

Response: this has been corrected as indeed it is useless to say so as discussed in Section XX.

Response: this has been corrected, there was incorrect referencing to the section

'upscale to larger scale.' - Redundant?

Response: indeed, this has been removed

"In what follows, we evaluate climatologies of evapotranspiration" - Maybe give some idea of what the reader should expect to learn?

Response: We have extended this sentence by: "in order to understand the typical seasonal cycles of those carbon, energy and water fluxes across the continental tropics."

Fig. 4 "over the wet part of the Amazon (top left), the Savanna region of Brazil (top right), " - Some readers may confuse the titles Amazon (wet) and Amazon (dry) for wet and dry seasons as opposed to regions. Perhaps replace with Amazon (rainforest) / Amazon (Savanna) or Amazon (rainforest) / Amazon (Cerrado), and then in the caption make the connection between those regions and wet vs dry climates.

Response: the figure legend has been corrected as indeed it was confusing

– Perhaps be more specific to ET components here "canopy evaporation (of inter- cepted rain)"

Response: this has been corrected as suggested

Fig. 5 Could you set the aspect ratio of the panels to make this less stretched out and easier to read? i.e., make the axes labels consistent and crop the ocean regions. I recommend stacking all the panels vertically so that panel 3 has the same longitude axis as the seasonal plots. That way we get a clear picture of how variable the SE Asian/Indonesian region is due to the topography (as noted in the text).

Response: yes we corrected this for all figures from Figure 5 to 10

Fig. 5 could be better integrated with the text - perhaps add references around line 367 'the topography and the distribution of island land masses leads to strong local variability [Fig. 5e]'

Response: yes we corrected, as suggested by the reviewer.

392-395 – "The seasonal pattern of ET resembles GPP..." - this section could use minor editing by breaking up the sentences and expanding to be more specific and clear.

Response: we have cut down this sentence and clarified it as: "Indeed, the seasonality reflects 1) the seasonality of water availability in drier, water-limited regions or 2) the seasonality of surface radiation in the wetter, more energy-limited portion of the Amazon."

"regions (Figure 5). GPP is maximized during the wet season in South America, as GPP is" - The text moves on to GPP without much transition here; perhaps add a transition sentence.

Response: we now better introduce the sentence by adding: "Integrated over the entire tropical latitudinal band, precipitation is highest in DJF and MAM when the wet season extends over most of the Amazon and adjacent savanna regions (**Error! Reference source not found.**). This seasonal cycle of precipitation largely determines the seasonal cycle of GPP. GPP"

It would help to define "moist static energy flux" as LH+SH

Response: indeed we have added LE+H now

I assume references for 'why do most contemporary land-surface models incorrectly represent the wettest rainforest GPP and ET...'' are in the prior sections? If so it may help to add a link here to refer readers back to the introduction.

Response: we have now added a link to section 3.2 discussing those challenges

You may mean that capturing this accurately will require better understanding?

Response: we removed this sentence that was not adding much but adding more confusion remove "because" in "because relative humidity is high..."

Response: this sentence has been corrected to: "water is typically not limiting for low-canopy species, as relative humidity is high and VPD is low"

"build up of water stress in the soil-plant continuum" - it may help to introduce the water potential terminology a bit earlier here, since it appears in the next paragraph anyway. That terminology may help to clarify this sentence.

Response: we have entirely rewritten this paragraph which was not clear.

Fix "also known as...

Response: corrected

Regarding midday depression, there are some references on this for tropical forests (Malhi et al., 1998; Williams et al., 1998; Harris et al., 2004).

Response: those references have been added

"We suggest that the most critical land-atmosphere feedbacks..." - It would help to specify this a bit, as to whether it is critical for understanding, addressing ESM water cycle deficiencies, modeling dynamic vegetation in a changing climate, etc. . .

Response: our focus here was to introduce the role of shallow clouds for this section. We have now toned down the sentence.

longwave cooling?

Response: this was not clear, we meant nighttime longwave cooling generates dew or forms dew

Response: we removed this part of the sentence which was confusing

- "As such the radiation feedback. . . may systematically impact clearings and de- forested regions". I suggest expanding and editing this sentence to reflect the three ideas it contains. The first is that transpiration is able to buffer the dry season effects in these regions, stabilizing ET, so that the feedback loop involving precipitation and ET is weakened. Thus, the impact of the dry season on ET (and hence clouds) is strongest in mesoscale clearings and deforested regions. In addition, the feedback of shallower clouds and surface radiation may be more important than the feedback of deeper clouds and precipitation.

Response: this is a good suggestion and has been corrected accordingly:" Indeed, higher transpiration in the dry season (due to the higher demand which is not entirely compensated by the slight water stress) can compensate the effect of reduced rain reevaporation intercepted by the canopy. As a result the feedback loop between precipitation and ET is weakened and the impact of the dry season on ET (and hence clouds) is strongest in mesoscale clearings and deforested regions (**Error! Reference source not found.**). In addition, the feedback of shallower clouds and surface radiation may be more important than the feedback of deeper clouds and precipitation."

- Here and in a few other places there is some discussion on respiration, in which the link to the water cycle may be lost for some readers as it is not as clear as for photosynthesis. Consider clarifying those connections.

Response: we have removed respiration wherever it could lead to confusion

"outweigh" 613 "increased lifetime" 618 "depending on the Amazonia site, from rather pristine..." ? 628 dynamics that drive

Response: those have been corrected

"the transition from turbulent clear convective conditions to shallow cloudy maybe modified in the future" - Do you mean changes in the frequency of the transition or nature of the transition?

Response: we have corrected this to "the frequency of the clear convective vs. shallow cloudy conditions may be modified in the future"

– the discussion on Maritime continent biomass burning is nice for geographical balance and is an outcome of precipitation deficits tied to El Nino. The carbon cycle impact of the burning is discussed in the review, but do you think it also has impacts on the water cycle that could be discussed here?

Response: as suggested by the reviewer we now include a discussion on the water cycle as well.

Figure 16 caption – could use a little more information on what increasing/decreasing regime mean

Response: this caption has been improved and extended

'reduced feedback strength' - reduced relative to what?

Response: indeed this was not sufficiently clear. We have now edited this sentence to clarify it.

References Malhi Y., A. D. Nobre, J. Grace, B. Kruijt, M. G. P. Pereira, A. Culf, S. Scott, Carbon dioxide transfer over a Central Amazonian rain forest. Journal of Geophysi- cal Research-Atmospheres 103, 31593-31612 (1998); published online EpubDec 27 (10.1029/98jd02647).

Harris P. P., C. Huntingford, P. M. Cox, J. H. C. Gash, Y. Malhi, Effect of soil moisture on canopy conductance of Amazonian rainforest. Agricultural and Forest Meteorology 122, 215-227 (2004); published online EpubApr 20 (10.1016/j.agrformet.2003.09.006).

Williams M., Y. Malhi, A. D. Nobre, E. B. Rastetter, J. Grace, M. G. P. Pereira, Seasonal variation in net carbon exchange and evapotranspiration in a Brazilian rain forest: a modelling analysis. Plant Cell and Environment 21, 953-968 (1998); published online EpubOct (10.1046/j.1365-3040.1998.00339.x).
As a review paper, there are not really any new findings here, but rather a summary of previous work. The paper does a fine job of this, and I believe it will be a valuable resource for others. I recommend acceptance with minor revisions.

Response: we thank the reviewer for the positive feedback

That being said, I do have some comments.

Lines 202-203: When I looked at the WECANN papers I did not see proof that seasonal cycles were reproduced. I see r-squared values in the tables, but not evidence showing seasonal cycles (also: I think the sentence should have 'variability' added at the end). I'm not sure I'm convinced that WECANN is better than other models in the tropics (really, Brazil. Do we have enough tower data in Africa or the Maritime Continent to really make an assessment?). Is there a way to establish this?

Response: we now add some caveats during the discussion to further emphasize that WECANN might have issues. However we point out that the retrieval better captured variability (mostly through the use of SIF than most other products).

This brings up something else: By using the words 'wet tropics', the implication is that tropical forests do not experience any water stress. I don't agree with this. There are a couple of papers [da Rocha et al., 2009, Costa et al., 2010] that discuss the variation in 'environmental control' (light limitation) and 'biotic control' (water limitation across precipitation and vegetation gradients in Brazil. Yes, the cerrado (savanna) is generally water-limited, and the wettest forest is light-limited, but the transition is not binary, nor is it limited to the cerradao (transition forest). I think a discussion of light- and water-limitation across gradients, and our uncertainty about the relative importance of each is an important part of tropical land-atmosphere interaction that is missing from this manuscript. George Vourlitis and coauthors have done some good work in the cerradao that should be described (I'm not going to list them all here). Baker et al. (2013) put forth a conceptual description of this gradient in a modeling study. Whether describing Brazil, the forest-to-desert transitions in Africa (both north and south), or the ecotone in Australia, there is important ecophysiological information in these transitions (and their response to changing climate) that is ignored here.

Response: we completely agree with the reviewer and indeed we see this as a continuum – we now have tried to clarify the text throughout.

On a potentially related note, I'm curious if the European Centre Amazon conversion papers need to be mentioned (Cox, Huntingford, Jones, et al.). I understand a review paper is not the same thing as a history paper, but these HADGCM papers got a lot of attention, and actually initiated quite a bit of investigation. The fact that there has been somewhat of a retreat from the initial findings means that these papers are no longer the 'state of the science', but they were seminal, along with the Saleska 2003 paper (in my opinion) in the initiation of some pretty important lines of investigation.

Response: those are good suggestions – we now have added those references which indeed are important.

**Figures:**

The figures don't flow smoothly with the text, in that the sequence of figures doesn't match when they are referenced. I was jumping back and forth in the figures as they were mentioned in the text, and I think a little rearranging would make the readability better.

Response: we have now corrected the ordering of the figures

Labels are too small in Figure 3, and are very difficult to read.

Response: those have been corrected and increased

Figures 5-10. The deformation of continents makes these plots hard to interpret. I would prefer to see the horizontal scale of the 4 seasonal plots stretched to match the longitude panel on the bottom of the plot. It would make the plots a little bigger, but readability would be improved. Also, the reader would be able to look directly up from the bottom panel and see the spatial variability in the latitudinal averages.

Response: we have now modified those plots as they were too stretched

Figure 16 seems to be thrown in, without much explanation in the text. I'm not sure I understand what is going on here, please clarify.

Response: we have clarified both the figure and caption as well as better described when it is referenced.

**Miscellaneous comments:**

Lines 74-75: Could include Friedlingstein (2006) and ?? here as citations.

Response: indeed this has been added

Lines 84-85: is it worth mentioning previous land-atm coupling papers, like Koster et al. (2004), or Dirmeyer (2011)?

Response: yes this was a big omission – those are added now

Lines 130-133: Cumbersome sentence, maybe some typos. Please reword.

Response: we have reformulated this to –

AMMA built upon previous field work in the region [*e.g. HAPEX-Sahel, Gourtorbe et al. 1993*]. This experiment advanced understanding of mesoscale convective systems and their initiation, as well as the role of surface processes

Line 169: If you say 15 S, you probably don't need the minus sign.

Response: corrected

Line 277: Looks like a laTex citation typo.

Response: corrected

Line 287: Grabowski (1999) could be cited too.

Response: this has been added

Line 289: 'models'

Response: this has been corrected

Line 324: I think the citation '[*Anber et al., 2015a*]' should be 'Anber et al. [2015a]', since the author name is part of the sentence. This looks like a place where \citep is used instead of \citet in the creation of the manuscript. There are a lot of instances of this in the manuscript, especially from line 587 on.

Response: those have been corrected throughout

Lines 398-400: 'Tropically-averaged EF does not evolve much...' With time? With space?

Response: we meant temporally, which is now added.

Lines 400-402: incomplete sentence.

Response: this has been corrected

Line 406: 'through' not 'though'

Response: corrected

Line 428: The text is talking about vertical gradients of light and water availability, yet refers to figure 3, which shows temperature response to ENSO. This recalls the earlier comment about figures and how they are referred to in the text.

Response: indeed, it was supposed to be figure 11, which has now been corrected.

Line 430: delete 'because'

Response: this sentence has been corrected to: However, water is typically not limiting for low-canopy species, as relative humidity is high and VPD is low, leading to low stress on understory conductance

Line 476: knownknows

Response: this has been corrected

Line 539: delete 'though,'

Response: this has been removed

Lines 549-550: I'm not sure I agree. Mesoscale-Induced clouds may be *initiated* preferentially in clearings and deforested regions, but they don't necessarily *stay* there. When I look at the GOES-16 images over Amazonia, I see clouds moving, not standing still. Do you have evidence that demonstrates that, integrated over time, the cleared/deforested regions are effected by clouds more than non-deforested regions? This may require some clarification.

Response: we have replaced this by saying that shallow clouds are triggered more. Yet, the life cycle of shallow clouds is very short (~30 minutes – i.e. PBL time scale) so that they cannot be advected very far, unlike mesoscale systems as we better calrify now.

Lines 567-569: confusing sentence, please reword.

Response: this has been reformulated

Lines 574-577: You might want to cite Fu and Li (2006) here as well.

Response: this has been added

Line 585: 'increased'

Line 625: 'lightning'

Lines 617-620: cumbersome sentences, some rewording would be helpful.

Line 703: inconsistent reference style.

Line 790: 'dry-to-wet'

Lines 815, 832: inconsistent reference style.

Line 893: 'through'

Response: these have been corrected

[revised manuscript text omitted]

$$\rho(x) = 1 - \exp\left(-\int_{x_0}^{x} \frac{E(x')}{C(x')u_x} dx'\right)$$
$$= 1 - \exp\left(-\frac{x}{\lambda(x)}\right),$$

Figure 16: Continental recycling ratio, $\rho$, and recycling length scale, $\lambda$, normalized by length scale, L, along the inflow direction x for atmospheric moisture either increasing or decreasing along the inflow path in the idealized model of Schäfli et al. (2012). $\rho$ represents how much of the rainfall is derived from terrestrial evapotranspiration, while $\lambda$ represents the length scale over which evapotranspired water is removed from the atmosphere via precipitation. Generally $\rho$ increases for larger distances into the continental interior, meaning recycling increases in importance, while $\lambda$ decreases. From Figure 8 of Schäfli et al. (2012).

[Figure]

[Figure]

Figure 17: 10 day back trajectory analysis over several regions of the continental tropics, along with LAI, mean TRMM estimated rainfall, and GLDAS ET estimates.

**Moved up [21]:** : Land-atmosphere feedback strength (change in the variance due to the feedback) between Precipitation and ET (top) and Photosynthetically Active Radiation (PAR) (bottom) based on recent metric developed by Green et al. [2017] using a multivariate Granger causality approach. ¶
¶

[Figure]

| Page 3: [1] Deleted | PIerre Gentine | 8/2/19 7:56:00 AM |
| --- | --- | --- |

| Page 3: [2] Deleted | PIerre Gentine | 8/2/19 7:56:00 AM |
| --- | --- | --- |

| Page 9: [3] Deleted | PIerre Gentine | 8/2/19 7:56:00 AM |
| --- | --- | --- |

| Page 9: [4] Deleted | PIerre Gentine | 8/2/19 7:56:00 AM |
| --- | --- | --- |

| Page 11: [5] Deleted | PIerre Gentine | 8/2/19 7:56:00 AM |
| --- | --- | --- |

| Page 11: [6] Formatted | PIerre Gentine | 8/2/19 7:56:00 AM |
| --- | --- | --- |

Comment Reference, Font: (Default) +Body (Calibri)

| Page 11: [7] Deleted | PIerre Gentine | 8/2/19 7:56:00 AM |
| --- | --- | --- |

| Page 11: [8] Deleted | PIerre Gentine | 8/2/19 7:56:00 AM |
| --- | --- | --- |

| Page 11: [9] Deleted | PIerre Gentine | 8/2/19 7:56:00 AM |
| --- | --- | --- |

| Page 11: [10] Deleted | PIerre Gentine | 8/2/19 7:56:00 AM |
| --- | --- | --- |

| Page 11: [11] Deleted | PIerre Gentine | 8/2/19 7:56:00 AM |
| --- | --- | --- |

| Page 11: [12] Deleted | PIerre Gentine | 8/2/19 7:56:00 AM |
| --- | --- | --- |

| Page 11: [13] Deleted | PIerre Gentine | 8/2/19 7:56:00 AM |
| --- | --- | --- |

| Page 11: [14] Deleted | PIerre Gentine | 8/2/19 7:56:00 AM |
| --- | --- | --- |

| Page 11: [15] Deleted | PIerre Gentine | 8/2/19 7:56:00 AM |
| --- | --- | --- |

| Page 11: [16] Deleted | PIerre Gentine | 8/2/19 7:56:00 AM |
| --- | --- | --- |

▼

| Page 11: [17] Formatted | PIerre Gentine | 8/2/19 7:56:00 AM |
| --- | --- | --- |

Comment Reference, Font: (Default) +Body (Calibri)

| Page 25: [18] Deleted | PIerre Gentine | 8/2/19 7:56:00 AM |
| --- | --- | --- |

▼

1.1

| Page 25: [19] Deleted | PIerre Gentine | 8/2/19 7:56:00 AM |
| --- | --- | --- |

▼

1.2

| Page 25: [20] Deleted | PIerre Gentine | 8/2/19 7:56:00 AM |
| --- | --- | --- |

▼

1.2.1

| Page 25: [21] Deleted | PIerre Gentine | 8/2/19 7:56:00 AM |
| --- | --- | --- |

▼

1.2.2

| Page 25: [22] Deleted | PIerre Gentine | 8/2/19 7:56:00 AM |
| --- | --- | --- |

▼

1.2.3

| Page 25: [23] Deleted | PIerre Gentine | 8/2/19 7:56:00 AM |
| --- | --- | --- |

▼

1.2.4

| Page 25: [24] Deleted | PIerre Gentine | 8/2/19 7:56:00 AM |
| --- | --- | --- |

▼

1.2.5

| Page 25: [25] Deleted | PIerre Gentine | 8/2/19 7:56:00 AM |
| --- | --- | --- |

▼

| Page 68: [26] Deleted | PIerre Gentine | 8/2/19 7:56:00 AM |
| --- | --- | --- |
▼
| Page 76: [27] Deleted | PIerre Gentine | 8/2/19 7:56:00 AM |

▼